



# Sensitivity of submarine melting on North East Greenland towards ocean forcing

Philipp Anhaus[1,2,*], Lars H. Smedsrud[1,2], Marius Årthun[1,2], and Fiammetta Straneo[3]

[1]Geophysical Institute, University of Bergen, Norway
[2]Bjerknes Centre for Climate Research, Bergen, Norway
[3]Scripps Institution of Oceanography, La Jolla, California, USA
[*]now at Alfred-Wegener-Institut Helmholtz-Zentrum für Polar- und Meeresforschung, 27570 Bremerhaven, Germany

**Correspondence:** Philipp Anhaus (philipp.anhaus@awi.de)

**Abstract.** The Nioghalvfjerdsbræ (79NG) is a floating ice tongue on Northeast Greenland draining a large part of the Greenland Ice Sheet. A CTD profile from a rift on the ice tongue close to the northern front shows that Atlantic Water (AW) is present in the cavity below, with maximum temperature of approximately $1\,°C$ at $610\,m$ depth. The AW present in the cavity thus has the potential to drive submarine melting along the ice base. Here, we simulate melt rates from the 79NG with a 1D numerical

Ice Shelf Water (ISW) plume model. A meltwater plume is initiated at the grounding line depth ($600\,m$) and rises along the ice base as a result of buoyancy contrast to the underlying AW. Ice melts as the plume entrains the warm AW. Maximum simulated melt rates are $50$ - $76\,m\,yr^{-1}$ within $10\,km$ of the grounding line. Within a zone of rapid decay between $10\,km$ and $20\,km$ melt rates drop to roughly $6\,m\,yr^{-1}$. Further downstream, melt rates are between $15\,m\,yr^{-1}$ and $6\,m\,yr^{-1}$. The melt-rate sensitivity to variations in AW temperatures is assessed by forcing the model with AW temperatures between $0.1$-$1.4\,°C$, as identified

from the ECCOv4 ocean state estimate. The melt rates increase linearly with rising AW temperature, ranging from $10\,m\,yr^{-1}$ to $21\,m\,yr^{-1}$ along the centerline. The corresponding freshwater flux ranges between $11\,km^3\,yr^{-1}$ ($0.4\,mSv$) and $30\,km^3\,yr^{-1}$ ($1.0\,mSv$), which is $5\,\%$ and $12\,\%$ of the total freshwater flux from the Greenland Ice Sheet since 1995, respectively. Our results improve the understanding of processes driving submarine melting of marine-terminating glaciers around Greenland, and its sensitivity to changing ocean conditions.

## 1 Introduction

With a total area of $1\,707\,400\,km^2$ and an ice volume of about $2\,850\,000\,km^3$, the Greenland Ice Sheet (GrIS) has a sea level rise potential of about $7\,m$ (Warrick et al., 1996; Reeh et al., 1999). In the past two decades the mass loss of the GrIS has quadrupled; from 1992-2000 to 2000-2011 the mass loss increased from $51\pm65\,km^3\,yr^{-1}$ to $211\pm37\,km^3\,yr^{-1}$ (Shepherd et al., 2012).

This accounts for about one quarter ($7.5\,mm$) of the total global sea level rise over this period (Church et al., 2011). The enhanced mass loss is caused by increased surface melt, and retreat and speed up of marine-terminating glaciers (Enderlin et al., 2014). The amount of freshwater that enters the ocean triggers changes in the estuarine circulation in the adjacent fjords (Straneo and Cenedese, 2015), the marine ecosystem (Cape et al., 2019), and atmospheric temperatures (Bamber et al., 2012). It





is also possible that the large scale ocean circulation might be impacted (Bamber et al., 2012; Straneo and Cenedese, 2015). The pathways through which the ice and freshwater are channeled and finally reach the ocean are the marine-terminating glaciers of the GrIS and the glacial fjords they discharge in, which are an important interface between the GrIS and the surrounding ocean. According to Enderlin et al. (2014), 15 of the largest glaciers are responsible for 50 % of the mass loss. Many of these

marine-terminating glaciers drain into fjords where they interact with the ocean (Straneo and Cenedese, 2015). Changes in the melting driven by Atlantic Water (AW) may cause glaciers to speed up and retreat (Straneo and Heimbach, 2013). The net mass loss is believed to be directly caused by the retreat of the glaciers due to melting at grounding line depth (Thomas et al., 2009). Submarine melting leads to an inward migration of the grounding line and an accelerated transport of the glaciers (Thomas et al., 2009). It is important to study submarine melting since it is a likely trigger of change of ice loss from the ice sheet.

The Nioghalvfjerdsbræ (79NG) located in Northeast Greenland is one of Greenland's major marine-terminating glaciers and one of three marine-terminating glaciers of the Northeast Greenland Ice Stream (NEGIS) (Fahnestock et al., 1993). Those glaciers together drain a large area of the GrIS (Fahnestock et al., 2001; Rignot and Kanagaratnam, 2006). The bedrock of the NEGIS is below sea level for about 150 km inland from the coast (Bamber et al., 2013; Thomas et al., 2009), making it potentially vulnerable to grounding-line retreat and loss in ice volume if subject to a reduction in buttressing at the terminus

(Dupont and Alley, 2005). Warm water between 0.5-0.8 °C around this area between 2006–2007 (Ingleby and Huddleston, 2007; Beszczynska-Möller et al., 2012) caused enhanced submarine melting and retreat of the grounding lines of the NEGIS glaciers (Khan et al., 2014). The 79NG is a 80 km long floating ice tongue that drains into Nioghalvfjerdsfjorden (e.g., Thomsen et al., 1997). Its terminus is split by an island into two glacier fronts (Thomsen et al., 1997). The main front is about 30 km wide and stretches eastwards, whereas a smaller front about 8 km wide terminates north in a neighboring narrow fjord termed

Dijmphna Sund (Mayer et al., 2000). Wilson et al. (2017) find that 79NG is subject to the highest melt rates of the remaining ice tongues in Greenland. Using a 3-D ice sheet model, Choi et al. (2017) found that 79NG remains stable in the future in response to ocean warming, even though melt rates increase strongly. In this study, submarine melting of the 79NG is investigated.

The analysis of Wilson and Straneo (2015) from CTD data collected in a rift on the ice tongue of 79NG in 2009 show evidence that AW present in the cavity below the ice tongue drives melting along the ice base. Both submarine melting and

subglacial discharge can drive an upwelling plume rising along the ice base due to a density contrast between the fresh meltwater and discharge and the AW. This has been observed at other sites (Jenkins et al., 2010; Straneo et al., 2010; Xu et al., 2012, 2013) and is likely to happen at the 79NG. The majority of the subglacial discharge is most likely released at depth and, as at many Greenland marine-terminating glaciers, probably stems mainly from surface runoff (Xu et al., 2013; Mankoff et al., 2016) draining the glaciers via a channelized network (Dallaston et al., 2015). Both, meltwater and glacial modified water were

found in the cavity of the 79NG (Wilson and Straneo, 2015). The buoyant plume can entrain AW and drive submarine melting (MacAyeal, 1985; Jenkins, 1991; Motyka et al., 2003). The submarine melting and subglacial discharge act as freshwater sources along the ice base (Jenkins, 1991, 2011). Approximately 50 % of the average annual freshwater release from the GrIS is associated with the calving of icebergs and the submarine melting (Bamber et al., 2012; Enderlin et al., 2014). Portions of the freshwater released at the 79NG likely drain into the East Greenland Current (EGC) and would then be transported further

south to areas where dense bottom waters are formed, e.g., Greenland Sea, Irminger Sea, and Labrador Sea, with the potential





to suppress this formation, which consequently would lead to changes in the Atlantic meridional overturning circulation (Curry and Mauritzen, 2005; Stouffer et al., 2006; Swingedouw et al., 2013; Rahmstorf et al., 2015). Understanding the variable submarine melting and associated freshwater flux is therefore important not only for the local ocean circulation (e.g. Yang et al., 2016).

The main goal of this study is to improve our understanding of the processes involved in submarine melting in order to asses the stability of the 79NG ice tongue. The focus lies on the evolution and dynamics of a buoyant plume and associated melt rates, and the sensitivity to AW temperature. The data and methods used to carry out this work are presented in Section 2. Results of hydrographic data, simulated melt rates from a 1D ISW plume model, and variability in melting due to variations in AW temperature and distribution in the water column are covered in Section 3. This is followed by a discussion in Section 4,

while Section 5 lists the conclusions and suggest future work to improve our understanding of ocean forced submarine melting.

## 2   Data and Methods

### 2.1   Topography and Bathymetry

Bedrock, ice elevation, and ice base are adopted from the Refined Topography 2 (RTopo2, Schaffer et al. (2016)). RTopo2 uses a variety of resolutions and projections, and observations were accordingly merged, interpolated, and smoothed. Specially, the

79NG ice base is derived from seismic, airborne, and radar surveys (Schaffer et al., 2016; Mayer et al., 2000). The ice elevation is calculated from the ice thickness exploiting hydrostatic equilibrium (Schaffer et al., 2016). Bedrock, ice elevation, and ice base along the centreline of the 79NG ice tongue are shown in Figure 1 and spatially in Figure 2b.

    Bathymetry observations are sparse on the continental shelf of Northeast Greenland, due to little ship traffic and extensive sea ice cover, so several data sets were merged. The poor data coverage is to some extent overcome by feeding a high-resolution

digital bathymetry model with reprocessed and combined multi- and single-beam echosoundings from more than two decades and maximum depths from CTD profiles (Arndt et al., 2015). Additionally GEBCO 2014 (General Bathymetric Chart of the Oceans; Weatherall et al., 2015) and IBCAOv3 (International Bathymetric Chart of the Arctic Ocean version 3) with a resolution of 500 m x 500 m were used.

    A realistic representation of the ice base is required for a proper simulation of the submarine melt rates and plume dynamics.

The horizontal resolution of RTopo2 is about 115 m × 615 m at 79.5 °N. Data points were manually extracted along the selected plume paths and smoothed by a moving average every 5th data point. The 1D ISW plume model uses a higher resolution for solving the equations, and requires a monotonically rising ice base along the path.

### 2.2   1D Ice Shelf Water plume model

A 1D Ice Shelf Water (ISW) plume model is used to simulate submarine melt rates and plume dynamics along the ice base

of 79NG ice tongue. The model was developed in MATLAB by Jenkins (1991). Frazil ice growth and precipitation as well as





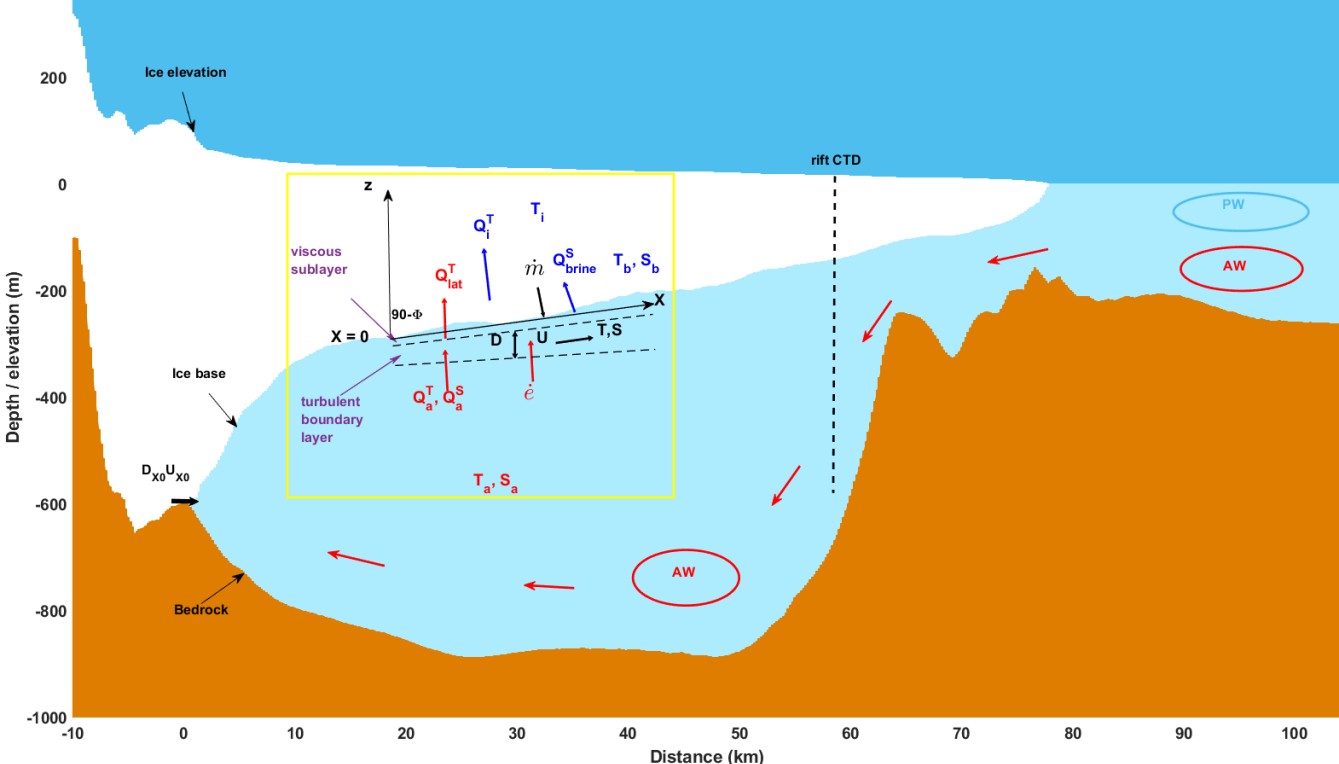

**Figure 1.** Bedrock, ice elevation, and ice base along the centreline profile of the 79NG ice tongue from RTopo2 (Schaffer et al., 2016). The inset box (yellow) illustrates an evolving plume circulation along the ice base modified from Jenkins (2011). Subglacial discharge $D_{X0}U_{X0}$ is released at the grounding line depth. The plume variables are defined in Section 2.2. The location of the rift CTDs and the location of the profile are marked in Figure 2b.

some tidal effects were included by Smedsrud and Jenkins (2004). The description of the model follows Jenkins (2011) unless otherwise stated.

Close to the grounding line the evolution of the plume is dominated by subglacial discharge rather than by submarine melting. The model is initialized with subglacial discharge (volume flux per unit width) which is assumed to be uniformly distributed along the grounding line. The STANDARD value used here is about $4.0 \times 10^{-3}\,\mathrm{m^2\,s^{-1}}$ (Table 1). It is derived from the long–term annual mean (1958–2017) of $74.75\,\mathrm{m^3\,s^{-1}}$ based on the RACMO2.3p2 surface runoff and then divided by the length of the grounding line which is roughly $20\,\mathrm{km}$.

The variables that describe the buoyant plume (Figure 1) are plume thickness D, velocity U, temperature T, and salinity S. Those are solved by four ordinary differential line source equations (ODEs) - one for each variable - that conserve fluxes of mass (Eq. 1), momentum (Eq. 2), heat (Eq. 3), and salt (Eq. 4) at the ice-ocean boundary:

$$\frac{d}{dX}(DU) = \dot{e} + \dot{m}, \qquad (1)$$





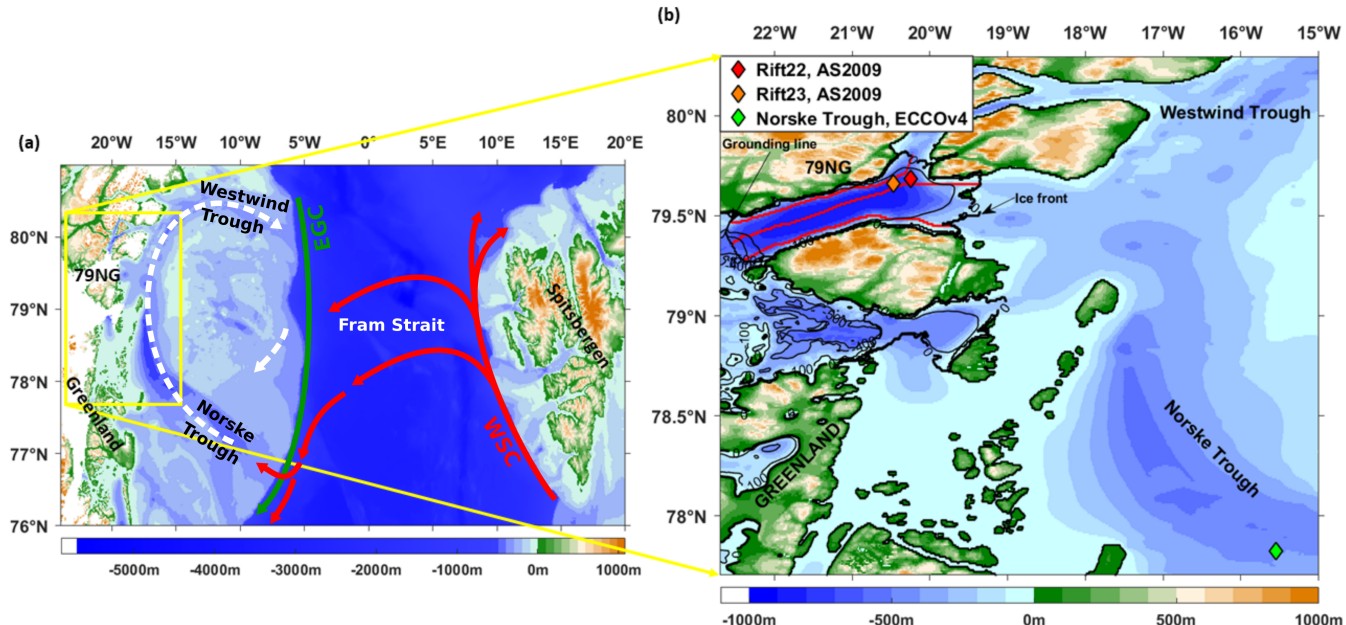

**Figure 2.** Bathymetry and topography in the Fram Strait and on the northeast Greenland shelf from RTopo2 (Schaffer et al., 2016). (a) Circulation relevant to the 79NG; schematics of the upper-layer circulation in the Fram Strait (Schauer et al., 2008) with re-circulating (Beszczynska-Möller et al., 2012; Hattermann et al., 2016) and northward Atlantic Water (AW, red) flow (Orvik and Niiler, 2002), the East Greenland Current (EGC, green, de Steur et al. (2009)), and the flow in the C-shaped trough system (white, Schaffer et al. (2017); Wilson and Straneo (2015); Budéus et al. (1997). Main currents are the West Spitsbergen Current (WSC) and the East Greenland Current (EGC), and the southern and northern recirculation in Fram Strait. (b) Zoom-in to the 79NG area. CTD stations are indicated by colored diamonds, together with water column thickness (blue shading). Contours show the ice base in 100 m intervals starting from 0 m at the ice front. Red lines along 79NG mark the pathways for the simulated ISW plumes (Section 2.2).

$$\frac{d}{dX}(DU^2) = Dg' \sin\phi - C_D U^2, \tag{2}$$

$$\frac{d}{dX}(DUT) = \dot{e}T_a + \dot{m}T_b - C_D^{1/2} U \Gamma_T (T - T_b), \tag{3}$$

$$\frac{d}{dX}(DUS) = \dot{e}S_a + \dot{m}S_b - C_D^{1/2} U \Gamma_S (S - S_b). \tag{4}$$

These four equations present the core of the model along X-axis distance, along the ice base starting at the grounding line (Figure 1.



The melt rate $\dot{m}$ is regulated by the entrainment $\dot{e}$ of ambient water, which here is normally AW (T > 0 °C, S > 34.3 psu). The entrainment has the physical unit thickness of water per unit time and is parameterized as $\dot{e} = \mathrm{E_0\,U\sin\phi}$, where $\mathrm{E_0}$ is the dimensionless entrainment coefficient and depends on the amount of turbulent mixing, and $\sin\phi$ is the slope of the ice base. The terms on the right-hand side of the momentum balance (Eq. 2) are the buoyancy force induced by the gravity component g acting

parallel to the ice base and the frictional drag parameterized by a dimensionless drag coefficient $\mathrm{C}_D$. g′ is the reduced gravity depending on the stratification. It is defined as $\mathrm{g'} = \mathrm{g}\left(\frac{\rho_p - \rho_a}{\rho_0}\right)$, with $\rho_p$ being the plume density, $\rho_a$ the density of AW, and $\rho_0$ a reference density. The plume density is described by a linear equation of state $\rho_p = \rho_0\left[1 + \beta(S - S_0) - \alpha(T - T_0)\right]$. Values for the thermal expansion $\alpha$ and haline contraction coefficient $\beta$ are given in Table 1. $T_0$ and $S_0$ are the initial temperature and salinity of the plume that must lie between the AW properties and the freezing point. Here, $T_0$ is set to the freezing point.

The initial plume velocity $\mathrm{U}_{X0}$ and plume thickness $\mathrm{D}_{X0}$ are calculated by assuming a balance between buoyancy and friction (Jenkins, 1991) and using the subglacial discharge. Melt rates at 79NG are expected to be highest at the grounding line. This is due to the pressure and salinity dependence of the freezing point $\mathrm{T}_b = \lambda_1\,\mathrm{S}_b + \lambda_2 + \lambda_3\,\mathrm{D}_e$, with $\lambda_3\,\mathrm{D}_e$ giving the ice base depth dependence. The ice-ocean boundary is symbolized by the subscript b. $\lambda_1, \lambda_2$, and $\lambda_3$ are constants presented together with other parameters used in Table 1. The heat needed to warm and subsequently melt the ice is provided by the AW. The

conservation of heat and salt at the ice-ocean boundary are expressed as $\mathrm{Q}_a^T = \mathrm{Q}_i^T$ - $\mathrm{Q}_{lat}^T$ and $\mathrm{Q}_a^S = \mathrm{Q}_i^S$ - $\mathrm{Q}_{brine}^S$ (Figure 1). $\mathrm{Q}_a^T$ and $\mathrm{Q}_a^S$ are the heat and salt fluxes associated with the AW. $\mathrm{Q}_i^T$ and $\mathrm{Q}_i^S$ are the heat and salt fluxes conducted into the ice, while $\mathrm{Q}_{lat}^T$ is the latent heat flux caused by melting or freezing, and $\mathrm{Q}_{brine}^S$ is the salt flux into the ice caused by melting.

The heat and salt fluxes at the ice base can be rewritten as

$$c\gamma_T(T - T_b) = \dot{m}c_i(T_b - T_i) + \dot{m}L, \tag{5}$$

and

$$\gamma_S(S - S_b) = \dot{m}(S_b - S_i), \tag{6}$$

where c and $c_i$ are the specific heat capacities for seawater and ice, L the latent heat of fusion which results from the transformation of solid ice to liquid, and $\gamma_T$ and $\gamma_S$ are the turbulent transfer coefficients for heat and salt. $\dot{m}$ $c_i$ ($T_b$ - $T_i$) is the heat directed into the ice. The ice temperature $T_i$ determines the amount of heat conducted and is set to -15 °C. The salinity of ice,

$S_i$, is always zero for melting.

It is important to differentiate between the turbulent boundary layer and the viscous sublayer (Figure 1). In the eddy dominated turbulent layer, heat and salt are diffused at same rates. In contrast, close to the ice-ocean boundary in the viscous sublayer, heat is exchanged much more rapidly than salt ($\gamma_T > \gamma_S$; Holland and Jenkins (1999); Straneo and Cenedese (2015)), accounted for in Eq. 7 and 8 by using a Prandtl ($Pr = 13.8$) and Schmidt ($Sc = 2432$) number. Those describe the ratio of the

kinematic viscosity of seawater $\nu$ to thermal and salinity diffusivities (Table 1). $\gamma_S$ is smaller than $\gamma_T$ by a factor 31.4 within the viscous sublayer, leading to melting being controlled by salt transfer (Jenkins and Bombosch, 1995).

$\gamma_T$ and $\gamma_S$ are velocity dependent and expressed as

$$\gamma_T = \frac{C_D^{1/2}U^2}{2.12\ln(C_D^{1/2}Re) + 12.5Pr^{2/3} - 9} \tag{7}$$





| Symbol | Value | Units | Description | Symbol | Units | Description |
|---|---|---|---|---|---|---|
| $E_0$ | $1.8 \times 10^{-2}$ | – | Entrainment coefficient | X | m | Distance along plume path |
| $C_D$ | $9.7 \times 10^{-3}$ | – | Drag coefficient | D | m | Plume thickness |
| $D_{X0}U_{X0}$ | $4.0 \times 10^{-3}$ | $m^2 s^{-1}$ | Subglacial discharge | U | $m s^{-1}$ | Plume velocity |
| $C_D^{1/2}\Gamma_{T,S}$ | $5.9 \times 10^{-4}$ | – | Stanton number | T | °C | Plume temperature |
| $C_D^{1/2}\Gamma_T$ | $1.1 \times 10^{-3}$ | – | Thermal Stanton number | S | psu | Plume salinity |
| $C_D^{1/2}\Gamma_S$ | $3.1 \times 10^{-5}$ | – | Haline Stanton number | $T_a$ | °C | AW temperature |
| $\lambda_1$ | $-5.73 \times 10^{-2}$ | $°C\,psu^{-1}$ | Seawater freezing point slope | $S_a$ | psu | AW salinity |
| $\lambda_2$ | $8.32 \times 10^{-2}$ | °C | Seawater freezing point offset | $T_b$ | °C | Temperature at ice-ocean boundary |
| $\lambda_3$ | $7.61 \times 10^{-4}$ | $°C\,m^{-1}$ | Depth dependence of freezing point | $S_b$ | psu | Salinity at ice-ocean boundary |
| L | $3.35 \times 10^5$ | $J\,kg^{-1}$ | Latent heat of fusion | $T_f$ | °C | In-situ freezing point of plume |
| $c_i$ | $2.009 \times 10^3$ | $J\,kg^{-1}\,K^{-1}$ | Specific heat capacity for ice | $\frac{\rho_p - \rho_a}{\rho_0}$ | – | Density contrast between plume and AW |
| c | $3.974 \times 10^3$ | $J\,kg^{-1}\,K^{-1}$ | Specific heat capacity for seawater | $\dot{m}$ | $m s^{-1}$ ($m\,yr^{-1}$) | Melt rate |
| $\alpha$ | $3.87 \times 10^{-5}$ | $K^{-1}$ | Thermal expansion coefficient | $\dot{e}$ | $m s^{-1}$ | Entrainment rate |
| $\beta$ | $7.86 \times 10^{-4}$ | – | Haline contraction coefficient | $\gamma_T$ | $m s^{-1}$ | Heat transfer coefficient |
| g | 9.81 | $m s^{-2}$ | Acceleration due to gravity | $\gamma_S$ | $m s^{-1}$ | Salt transfer coefficient |
| $\rho_i$ | 917 | $kg\,m^{-3}$ | Average density of ice | $D_e$ | m | Depth of ice base |
| $\rho_w$ | 1024 | $kg\,m^{-3}$ | Average density of water | $\sin\phi$ | – | Slope of ice base |
| Pr | 13.8 | | Prandtl number | | | |
| Sc | 2432 | | Schmidt number | | | |
| $\nu$ | $1.95 \times 10^{-6}$ | $m^2 s^{-1}$ | Kinematic viscosity of sea water | | | |

**Table 1.** Physical constants (left) and output variables (right) of the 1D ISW plume model.

and

$$\gamma_S = \frac{C_D^{1/2}U^2}{2.12\ln(C_D^{1/2}Re) + 12.5Sc^{2/3} - 9} \tag{8}$$

adopted from Kader and Yaglom (1972, 1977). Re is the Reynolds number.

The width of the fjord gives an indication for if rotational effects, and, hence, across-fjord variations should be considered. The fjord width of 20 - 30 km exceeds the first baroclinic radius of deformation being about 9 km calculated as $R = \left(\frac{g'H_1H_2}{H}\right)^{0.5} \cdot f^{-1}$. As a consequence, rotation effects the dynamics in the fjord. $H_1$ (90 m), $H_2$ (470 m), and H (560 m) are the upper, lower, and total water column thickness provided from the CTD profile in the rift on the 79NG ice tongue. The stratification is assumed to be a two-layer system consisting of PW overlying AW with $\rho_1 = 1026.3\,kg\,m^{-3}$ and $\rho_2 = 1027.7\,kg\,m^{-3}$. f is the Coriolis parameter. The 1D the model is not capable of representing across-fjord variations directly and solves the problem along predefined flow lines (Figure 2b, red lines). The differences between the different flow paths are modest, and the essential spatial variability remains along the X-axis, guided by the southern coast.

The model is steady in time, uniform in the across-flow direction, and depth-integrated. The ODEs are solved by $4^{th}$ and $5^{th}$ order Runge-Kutta formulas. Inputs are calculations along plume gradients of the prognostic variables D, U, T, and S. The integration stops when the plume has reached its level of neutral buoyancy. The output data are vectors of derivatives D, U, T,





and S along the ice base. Subsequently, the melt rate and associated variables are calculated according to Eq. 5 and 6. Input variables are D, U, T, S, and the depth of the ice base $D_e$. Melting occurs when the plume temperature is above its freezing temperature with S instead of $S_b$. The simulated variables are presented in Table 1 (right).

## 2.3  Atlantic Water temperatures and subglacial discharge

To assess the sensitivity of submarine melting toward variable AW forcing we use temperature and salinity from ECCOv4 (Estimating the Circulation and Climate of the Ocean version 4 release 3). This is is an observational constrained and dynamically consistent ocean state estimate covering the time period 1992-2015 (Forget et al., 2015). ECCOv4 is generated by the Massachusetts Institute of Technology general circulation model (MITgcm) on a horizontal grid of $1°$ and with 50 layers in the vertical, with the observational constraint provided by satellite data, Argo floats, hydrographic profiles, and moorings.

To achieve an estimate of the subglacial discharge that drains into the cavity underneath the 79NG, simulated daily surface runoff from RACMO2.3p2 (Regional Atmospheric Climate Model; Noël et al. (2018)) with a 1 km x 1 km resolution is utilized. The surface runoff is the fraction of melt and rainfall that does not refreeze and percolates to the bed and along the surface of the ice sheet and eventually drains into the fjord. The long-term annual mean subglacial discharge value for the period 1958–2016 is used here. This was calculated doing flow routing of surface runoff based on the 79NG bed and surface topography over the
full catchment area (10 to 40°W, 75 to 80°N), and equally distributing the flux along the grounding line.

## 3  Results

### 3.1  Hydrography - Observations and Ocean State Estimate

AW recirculates in Fram Strait and flows onto the continental shelf of Northeast Greenland (Hattermann et al., 2016) (Figure 2a). A C-shaped trough system exists in front of 79NG (Figure 2a), which is divided into the southern Norske Trough and the
northern Westwind Trough (Wilson and Straneo, 2015). The main aim of this section is to investigate seasonal and interannual variations in AW properties close to 79NG. The identified range in AW properties from both observations and model simulations will be implemented in the 1D ISW plume model (Section 3.3) in order to quantify the associated submarine melt rate variability.

Observations show that the AW in the Norske Trough flows toward the main front of the 79NG ice tongue within a boundary
current along the northeastern slope of the trough and into the cavity beneath the ice tongue (Wilson and Straneo, 2015; Schaffer et al., 2017). During a survey in September 2009 with M/V Arctic Sunrise (AS2009 hereafter) CTD profiles were taken in a rift on the ice tongue, 15 km upglacier from the northern terminus (Wilson and Straneo, 2015). The data reveal the presence of AW as warm as $1°C$ at depths in the cavity (Figure 3, red and orange diamonds). The AW underlies a shallow layer of cold and fresh Polar Water (PW; $T < 0°C$, $S < 34.3$ psu), which originates in the Arctic Ocean and is carried onto the continental shelf
by the East Greenland Current (Mayer et al., 2000; Wilson and Straneo, 2015; Schaffer et al., 2017). The AW is found below

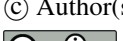



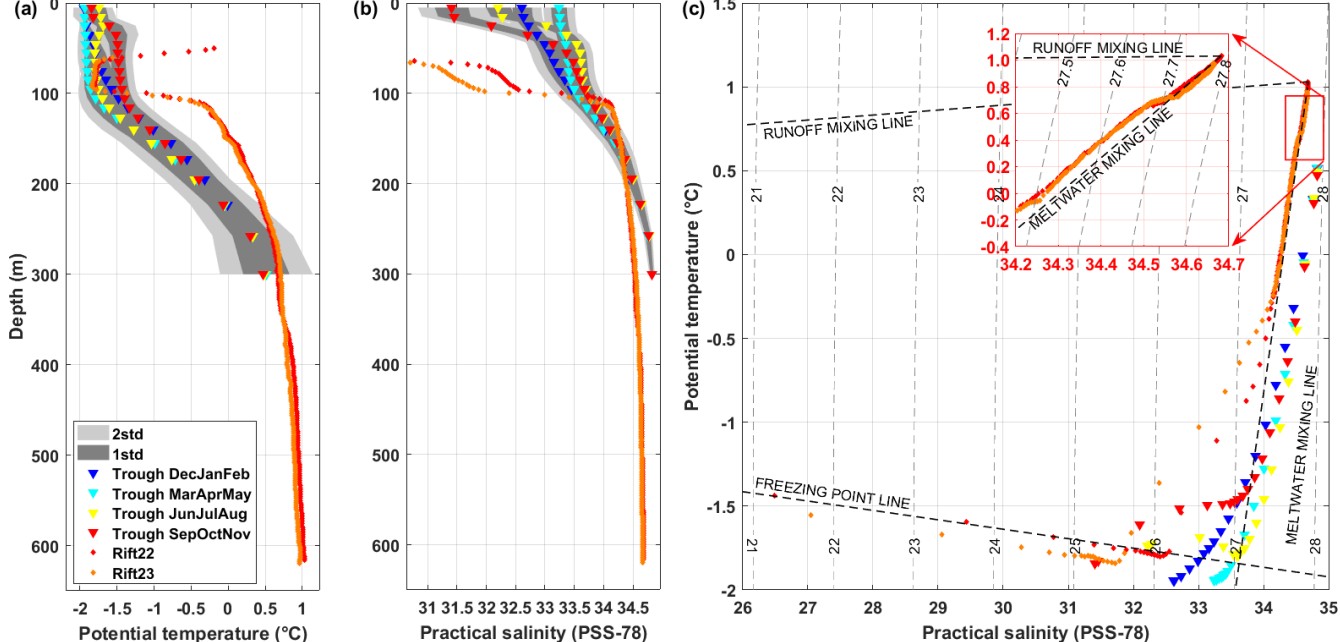

**Figure 3.** (a) Potential temperature and (b) practical salinity as a function of depth in the rift (red and orange diamonds, Wilson and Straneo (2015)) and in the Norske Trough (colored triangles, ECCOv4). The ECCOv4 profiles are seasonal means between 1992–2015. The gray shadings indicate the (dark) standard and the (light) 2 times standard deviations. The location of the profiles is shown in Figure 2b. (c) Potential temperature versus practical salinity diagram for the same profiles. Density ($\sigma_0 = \rho_0$ - $1000\,\mathrm{kg\,m^{-3}}$) lines are drawn every $1\,\mathrm{kg\,m^{-3}}$. The freezing point temperature referred to the surface pressure is indicated by the black dashed line. The meltwater mixing line connects the densest and warmest AW found in the rift profiles (T = $1.0\,^\circ$C, S = $34.68\,\mathrm{psu}$) and the effective ice properties ($\mathrm{T}^* = $ -$92.8\,^\circ$C, $\mathrm{S}_i = 0\,\mathrm{psu}$). The effective ice temperature is $\mathrm{T}_i^* = \mathrm{T}_f - \frac{L}{c} - \frac{c_i}{c}$ ($\mathrm{T}_f$ - $\mathrm{T}_i$). The mixture of ambient water and glacial runoff fall along the runoff mixing line with the runoff temperature and salinity being T = $0\,^\circ$C and S = $0\,\mathrm{psu}$ (Straneo et al., 2012). The inset figure shows an expanded view of the deepest water masses.

150 m, and the hydrographic properties are very similar to those observed in the Norske Trough, supporting a flow of AW into the cavity (Wilson and Straneo, 2015).

Evidence of submarine melting within the sub-ice shelf cavity is provided by the presence of glacial meltwater-modified water found in the observed rift profile (Wilson and Straneo, 2015). This water mass is identified by the meltwater mixing line

5 that connects the warmest AW with the effective ice properties (Figure 3c). The runoff mixing line furthermore connects AW with surface runoff and/or subglacial discharge. The water in the cavity veers off against the runoff mixing line (Figure 3c inlet), suggesting the presence of a mixture between AW and surface runoff and/or subglacial discharge, referred to as glacial runoff-modified water (Wilson and Straneo, 2015).

Mean seasonal temperature and salinity profiles in the Norske Trough for the period 1992–2015 from ECCOv4 are shown

10 in Figure 3 (colored triangles). Simulated AW is, on average, found at depths below 225 m. PW occupies most of the water



column down to 195 m. Between 50 m and 100 m the water column is very cold and well mixed in temperature (Figure 3a, less pronounced during December-February). In contrast, the salinity increases steadily with depth (Figure 3b). Temperature and salinity increase gradually from 100 m to their maximum values at 300 m depth.

The annual mean AW temperature at 300 m depth varies between 0.1 °C and 1.4 °C (not shown). The upper extension of AW

in the water column furthermore varies by 65 m; from 195 m to 260 m. Seasonal variations in AW temperature are in the mean not significant (Figure 3a, b). Both seasonally and inter-annually, the AW salinity variability is small; ranging from 34.78 psu to 34.87 psu (Figure 3b). The mean AW temperature and salinity at 300 m are 0.5 °C and 34.82 psu, respectively (Figure 3a, b, dark gray shading). The very cold and well mixed layer subsurface prevents heat loss of AW towards the atmosphere. As a consequence, we assume that AW simulated by ECCOv4 does not cool significantly when it flows from the Norske Trough

into the cavity beneath the 79NG, and the temperature range simulated by ECCOv4 is applied in the plume model underneath the ice tongue.

## 3.2   1D Ice Shelf Water plume model - standard case

The 1D numerical ISW plume model has been applied to three flow lines along the ice base of the 79NG ice tongue (Figure 2b, red lines). The evolution of the ice base as a function of the along-tongue direction for the three line is shown in figure

8. The hydrographic properties of the water column in the cavity below the ice tongue are provided by the CTD rift profile 23 (Figure 3a and b, orange). As the AW is present below 150 m, the ice tongue is in direct contact with AW (Figure 1). We focus on the evolution of submarine melt rates and plume dynamics with respect to the distance from the grounding line. First, results obtained using optimal model parameters for the 79NG are described, referred to as the STANDARD case (Section 3.2). Afterwards sensitivity experiments of the submarine melt rate to the AW properties are presented, including spatial variability

(Section 3.3). Then, the influence of the entrainment coefficient, the drag coefficient, and the subglacial discharge on the melting are investigated (Section 3.4). Lastly, a simplified method is used to yield estimates of the stability of 79NG, time scale of disappearance, and the fresh water export associated with the melting (Section 3.5).

In the STANDARD case, the model was applied to the centreline of the 79NG ice tongue (Figure 1, 2b and 8e). The chosen set of parameters used for the STANDARD case are summarized in Table 2. The explanation for these chosen model

| Parameter | Symbol | Value | Unit | Equation / Figure | Reference |
|---|---|---|---|---|---|
| Ice base | − | − | m | Figure 1 | RTopo2, Schaffer et al. (2016) |
| Ice base slope | $\sin\phi$ | − | − | − | RTopo2, Schaffer et al. (2016) |
| CTD profile | − | − | − | Figure 3 | AS2009, Wilson and Straneo (2015) |
| Entrainment coefficient | $E_0$ | $1.8 \times 10^{-2}$ | − | $\dot{e} = E_0 \, U \sin\phi$ | following discussions by Pedersen (1980) and Jenkins (1991) |
| Drag coefficient | $C_D$ | $9.7 \times 10^{-3}$ | − | − | Jenkins et al. (2010) |
| Subglacial discharge | $D_{X0} U_{X0}$ | $4.0 \times 10^{-3}$ | $\mathrm{m^2 \, s^{-1}}$ | − | RACMO2.3p2, Noël et al. (2018) |
| Transfer coefficients | $\gamma_{T,S}$ | − | $\mathrm{m \, s^{-1}}$ | Eq. 7 and 8 | Jenkins (1991) |

**Table 2.** Model parameters used in the STANDARD case.







**Figure 4.** Model results along the centreline of 79NG in the STANDARD case (Table 2). (a) Submarine melt rate, (b) density contrast between AW and the plume, (c) thermal drivings, (d) plume temperature, (e) plume salinity, (f) plume velocity, and (g) plume thickness. GLZ indicates Grounding Line Zone, TZ Transition Zone, and DSZ Down-Stream Zone.

parameters, as well as a short description of their influences on the submarine melt rates and the plume dynamics, are given at the end. The overall results is that the ice base changes laterally and determines a spatially varying distribution of the melt rate and plume dynamics. The plume starts at the Grounding Line (GL, X = 0 km) and continues along the ice base until it is no longer buoyant at about X = 75 km (Figure 4).

5    The plume quickly entrains AW causing the temperature and salinity to rise to a maximum within the first 10 km - the Grounding Line Zone (GLZ). This in turn causes the maximum melt rates between 50 - 76 m yr$^{-1}$ (Figure 4). AW at the GL depth is about 3 °C warmer than the in situ $T_f$, which explains the large thermal driving in Figure 4c. The plume itself has a




maximum temperature between -0.5 to -1.0 °C, and is gradually approaching -2.0 °C (Figure 4d). The $T_a$, $S_a$ and the melting thus have the same shape along the plume path with a maximum within the GLZ. In contrast is the density contrast between the plume and surrounding AW quite constant, producing a typical plume speed of $10\,\mathrm{cm\,s^{-1}}$ and a total traveling time from the grounding line to the main front of 8.5 days (Figure 4b and f).

The plume is driven by a density contrast $\rho_a$ - $\rho$ between AW and the plume itself. The density contrast is strongest close to the GL (Figure 4b). This is because of the fresh subglacial discharge released at this position and providing the initial buoyancy for the plume. The density contrast decreases rapidly because as the plume accelerates and its velocity increases (Figure 4f) more AW is entrained. Moreover, the slope is steepest within the GLZ (Figure 8e) and, thus, entrainment of AW is high. However, as freshwater is released into the plume due to melting, the density contrast increases again until about 30 km

downstream. The increase in the amount of freshwater is evident in the salinity decrease along the ice base (Figure 4e). Down-glacier of 30 km, the density contrast continuously decreases as a result of less melting (Figure 4a). At about 75 km the plume reaches its level of neutral buoyancy (Figure 4b and Table 3).

The temperature evolution of the plume compared to the surroundings changes along the flow path (Figure 4c). The available heat source is given by the AW temperature and how much warmer this is above the in-situ freezing point at the ice-ocean

boundary ($T_a$-$T_f$). This thermal driving is highest at the GL with about 3.3 °C, decreasing nearly continuously along the ice

| Model run | Mean melt rate ($\mathrm{m\,yr^{-1}}$) | Maximum melt rate ($\mathrm{m\,yr^{-1}}$) | Point of detachment (km) | Final plume flux ($\mathrm{m^3\,s^{-1}}$) | Travel time (days) |
|---|---|---|---|---|---|
| STANDARD | 15.2 | 76.4 | 75.3 | 38 918 | 8.5 |
| 50 % ($E_0 = 0.9 \times 10^{-2}$) | 6.3 | 41.9 | 75.5 | 15 000 | 12.8 |
| 200 % ($E_0 = 3.6 \times 10^{-2}$), 90 % ($C_D = 8.6 \times 10^{-3}$) | 31.5 | 113.4 | 75.5 | 94 200 | 6.3 |
| 25 % ($C_D = 2.5 \times 10^{-3}$) | 21.6 | 75.1 | 69.7 | 60 000 | 4.2 |
| 50 % ($C_D = 4.9 \times 10^{-3}$) | 17.7 | 77.9 | 74.5 | 48 900 | 6.2 |
| 90 % ($C_D = 8.6 \times 10^{-3}$) | 15.5 | 76.1 | 75.1 | 39 900 | 8.1 |
| 200 % ($C_D = 1.9 \times 10^{-2}$) | 11.3 | 68.4 | 78.3 | 29 100 | 13.8 |
| $D_{X0}U_{X0} = 4.0 \times 10^{-4}\,\mathrm{m^2\,s^{-1}}$ | 14.6 | 75.0 | 75.3 | 37 897 | 8.7 |
| $D_{X0}U_{X0} = 4.0 \times 10^{-2}\,\mathrm{m^2\,s^{-1}}$ | 16.5 | 82.5 | 75.3 | 43 869 | 7.9 |
| $D_{X0}U_{X0} = 7.5 \times 10^{-2}\,\mathrm{m^2\,s^{-1}}$ | 17.3 | 85.9 | 75.3 | 47 261 | 7.6 |
| South coast | 11.6 | 50.7 | 60.4 | 27 600 | 7.9 |
| North coast | 1.3 | 20.4 | 77.8 | 45 510 | 41.6 |
| +0.5 °C | 18.9 | 94.5 | 75.3 | 41 624 | 7.8 |
| warm AW | 20.5 | 101.2 | 75.3 | 42 745 | 7.6 |
| cold AW | 10.2 | 53.7 | 75.3 | 33 339 | 10.2 |
| shallow AW | 15.2 | 75.7 | 75.3 | 38 162 | 8.6 |
| deep AW | 13.3 | 72.7 | 75.3 | 37 681 | 8.7 |

**Table 3.** Sensitivity of the model results along the centreline of the 79NG ice tongue due to entrainment coefficient, drag coefficient, subglacial discharge, ice bases, and AW properties.





base to $1.7\,^\circ$C close to the main front. A core assumption in the model is that the ice-ocean boundary is at the in-situ freezing point. Hence, a positive $T_a$-$T_f$ means melting. Although $T_a$-$T_f$ decreases along the flow path, it maintains a high positive value explaining the melting (Figure 4a). In contrast, the thermal driving between the plume and the in-situ freezing point, T-$T_f$, decreases along the flow path consistent with T adjusting toward $T_f$. $T_f$ increases steadily because the plume reaches

shallower depths and freshens due to melting (Figure 4c and e). The same is simulated for the ice-ocean boundary temperature $T_b$ (Figure 4c, red), although it is higher within the GLZ because $S_b$ is used instead of the plume salinity S to calculate $T_b$. Further downstream, $T_b$ and $T_f$ are almost identical due to the decrease in S (Figure 4e).

The plume accelerates linearly within $5\,$km due to a density contrast initiated by the release of subglacial discharge and further downstream due to melting. It reaches a maximum velocity there of about $19\,\mathrm{cm\,s^{-1}}$ (Figure 4f). Besides three local

minima where the plume velocity rapidly decreases the plume maintains a fairly high mean velocity of about $10\,\mathrm{cm\,s^{-1}}$. However, at approximately $70\,$km, the velocity decreases again to below $5\,\mathrm{cm\,s^{-1}}$. Small scale variability is due to the ice base. The plume thickness D increases nearly linearly from $0\,$m at the GL to about $17\,$m within $20\,$km (Figure 4g). Downstream from that location D experiences a sudden drop to $6$-$7\,$m within $1\,$km. This pattern is observed two more times along the ice base where D rises to about $18\,$m and $14\,$m. These drops are consistent with an increase in the plume velocity (Figure 4f). However,

the mean thickness remains rather constant of about $10\,$m until close to $70\,$km (Figure 4g). The final plume flux, a product of the final plume velocity and thickness and the width of the main front ($30\,$km) along which it would be released in a 2D or 3D concept, is about $38\,900\,\mathrm{m^3\,s^{-1}}$ (Table 3).

The entrainment coefficient $E_0$ governs the entrainment rate $\dot{e}$ and the amount of AW that is transferred into the plume, and the value applied here is $1.8\,\mathrm{x}\,10^{-2}$. The drag coefficient $C_D$ is important for the flow along the ice base, especially for the

plume velocity, and the value $9.7\,\mathrm{x}\,10^{-3}$ is adopted from Jenkins et al. (2010). The subglacial discharge $D_{X0}U_{X0}$ is the source for the plume and makes it rise initially. The long–term annual mean $4.0\,\mathrm{x}\,10^{-3}\,\mathrm{m^2\,s^{-1}}$ simulated using RAMCO2.3p2 (Noël et al., 2018; Todd et al., 2018) is used here.

### 3.3 Ice Shelf Water plume - ocean forcing sensitivity

The main question investigated here is the response of submarine melt rates on 79 NG due to a warming ocean, but we include

a full temperature range of the ocean forcing. At the end melting variability across the ice tongue is assessed by considering three flow lines with different ice bases.

Since the 1950s there has been a wide spread warming throughout the subpolar North Atlantic (Straneo and Heimbach, 2013) and the Arctic/sub-Arctic Ocean (Beszczynska-Möller et al., 2012; Onarheim et al., 2014; Polyakov et al., 2017), including the Norske Trough (Schaffer et al., 2017). As a result of this warming, the submarine melting of many marine-terminating glaciers

in Greenland have likely intensified (Straneo and Heimbach, 2013). To assess the overall sensitivity to such a warming, the AW temperature in the cavity (rift profile) was increased by $0.5\,^\circ$C. This increases the mean melting by 24% from $15\,\mathrm{m\,yr^{-1}}$ in the STANDARD case to to $19\,\mathrm{m\,yr^{-1}}$ (Figure 5 and Table 3).

A stable ice tongue means $m_{sub} + m_{sfc} - \mathrm{ice}_{flux} = 0$, where $m_{sub}$ is the mean submarine melt rate, $m_{sfc}$ the mean surface melting, and $\mathrm{ice}_{flux}$ the inflowing ice flux from up-glacier. Wilson et al. (2017) estimated $10.2\pm0.59\,\mathrm{km^3\,yr^{-1}}$ for $\mathrm{ice}_{flux}$





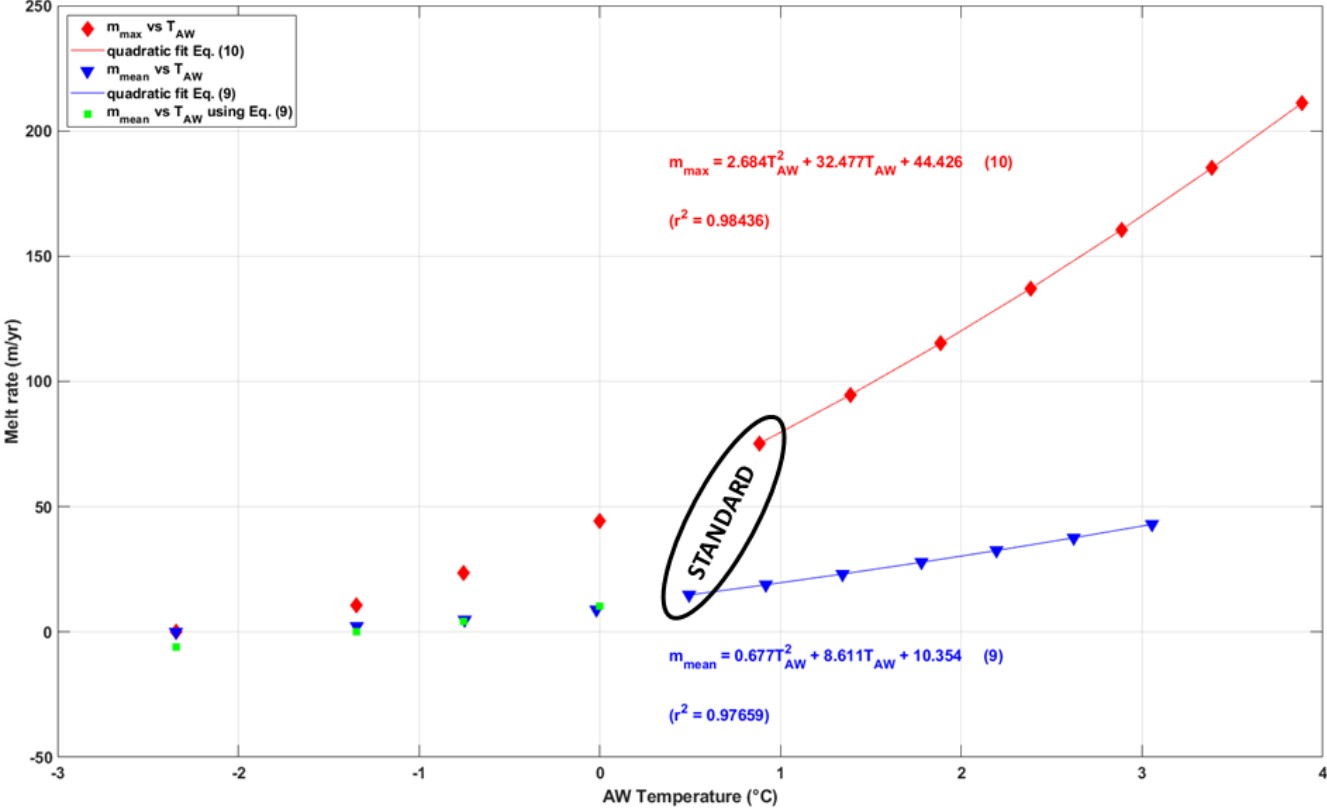

**Figure 5.** Mean melt rate along the centreline of the 79NG ice tongue (blue), maximum melt rate in the grounding line zone (red), and melt rate calculated using the quadratic fit function (green) depending on the AW temperature as described in the text.

and $2.3 \pm 1.3\,\mathrm{km^3\,yr^{-1}}$ for $\mathrm{m}_{sfc}$. Applied to the surface geometry of the 79NG ice tongue (Section 3.5), this converts to about $5.5\,\mathrm{m\,yr^{-1}}$ and $1.2\,\mathrm{m\,yr^{-1}}$, respectively. This results in a submarine melting of about $4.2\,\mathrm{m\,yr^{-1}}$ required for the 79NG to be stable. The equation for the freezing point temperature from McDougall and Barker (2011) yields approximately -2.3 °C for a salinity of 34.66 psu and at 600 m depth (Figure 3b). This is the lower bound in temperature, and the ISW plume model

5 produces no melting, i.e. freezing. Warmer temperature than -2.3 °C leads to more melting and the simulated melt rates (mean and maximum; Figure 5) increase almost linearly with ocean temperature (Eq. 9 and 10). The mean melt rates calculated using Eq. 9 match the model results well above -1.3 °C. Note that for a stable ice tongue ($\mathrm{m}_{sub} = 4.2\,\mathrm{m\,yr^{-1}}$), equation 9 yields a AW temperature of approximately -0.8 °C. However, observations in the last decades show significantly warmer temperatures (Mayer et al., 2018; Schaffer et al., 2017; Wilson and Straneo, 2015; Mayer et al., 2000)

10  $$m_{mean} = 0.677 \cdot T_{AW}^2 + 8.611 \cdot T_{AW} + 10.354 \qquad (9)$$





and

$$m_{max} = 2.684 \cdot T_{AW}^2 + 32.477 \cdot T_{AW} + 44.426. \tag{10}$$

**Atlantic Water forcing.** The response of the submarine melt rates to variations in AW properties (Figure 3) in the Norske Trough between 1992 and 2015 from the ECCOv4 ocean state estimate is now investigated. We assume that the properties in the Norske Trough are applicable to the hydrography in the cavity beneath the 79NG ice tongue because ECCOv4 does not have an ice tongue. The AW temperature at 300 m ranges from 0.1 °C (cold AW) to 1.4 °C (warm AW). The upper depth of AW varies from 195 m (shallow AW) to 260 m (deep AW). Based on this range in AW properties, the observed hydrography in the cavity provided by the CTD profile taken in the rift was adjusted accordingly (Figure 6). Variations in AW salinity are not significant, and are thus not examined here.

As a response to AW temperature changes, the 1D ISW model simulates mean submarine melt rates between $10 \, \mathrm{m \, yr^{-1}}$ and $21 \, \mathrm{m \, yr^{-1}}$ (Table 3). Maximum melting occurs within the GLZ, varying between $55 \, \mathrm{m \, yr^{-1}}$ and $101 \, \mathrm{m \, yr^{-1}}$. In the warm AW case (red) melting increases everywhere along the ice base (Figure 7). The increase is most pronounced in the GLZ. The mean melt rate increases to about $21 \, \mathrm{m \, yr^{-1}}$ (Table 3). In the cold AW case (blue) melting decreases, strongest within the GLZ. The mean melt rate decreases to about $10 \, \mathrm{m \, yr^{-1}}$. Bringing AW closer to the ice base (shallow AW, green) and deeper into the water column (deep AW, orange) alters the melting not significantly within 50 km (Figure 7 and Table 3). Further downstream, melting changes slightly with an increase in the shallow AW case and a decrease in the deep AW case.

**Spatial Variability.** The ice base of the 79NG varies not only along its ice tongue, but also changes across it (Figures 2 and 8e). The thickest ice base is concentrated along the centreline consistent with the fastest ice flow (Mouginot et al., 2015). Towards the margins the ice base thins due to a higher lateral drag along the margins that counteracts the ice flow (Benn and Evans, 2010). The spatial distribution of the ice base influences the submarine melt rates. The steeper the ice base the faster the plume and, thus, the higher the entrainment of AW and the melting. Along the north coast (Figures 2 and 8e) melt rates are much lower (near zero) compared to the center (Figure 8d). The melt rates along the south coast are slightly lower compared to the center (Figure 8d). The spatial variability both longitudinal and transverse is similar to results obtained by Wilson et al. (2017) with the melt rates along the south coast matching best.

### 3.4 Ice Shelf Water plume - parametrization

Our results are not invariant to choices of model parameters within the ISW plume model. In this section we assess the sensitivity to the most important; the entrainment coefficient, the drag coefficient, and the subglacial discharge. Overall, we find that the results are robust to the large ranges tested for below.

**Entrainment Coefficient.** The entrainment rate $\dot{e}$ regulates the amount of AW put into the plume and, thus, the heat available for melting. The heat supplied to the ice-ocean boundary is proportional to both the AW temperature and the entrainment rate (Eq. 3) (Jenkins, 1991). Increasing the entrainment coefficient $E_0$ leads to an increase in $\dot{e}$ and, hence, in the heat supply as shown in Figure 8a. $E_0$ was increased to $200\,\%$ ($E_0 = 3.6 \times 10^{-2}$; Jenkins, 1991) of the STANDARD value ($100\,\%$, $E_0 = 1.8 \times 10^{-2}$). This results in elevated melt rates everywhere along the ice base, in particular within the GLZ and the TZ





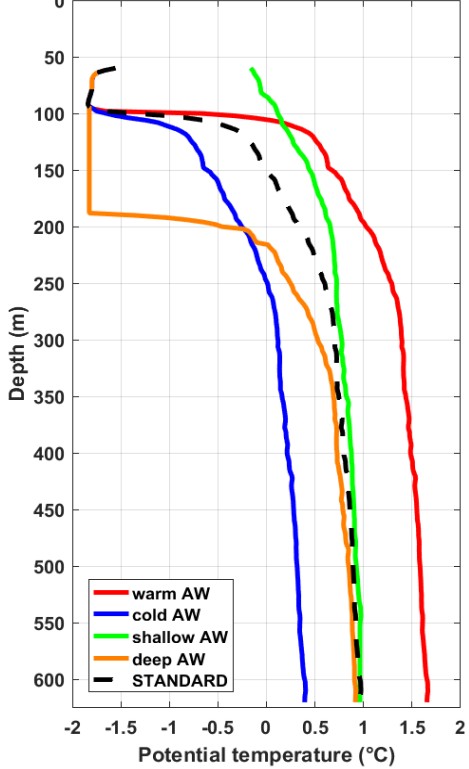

**Figure 6.** Potential temperature in the cavity below the 79NG ice tongue from the rift profile close to the northern front (black dashed). Warm AW (red), cold AW (blue), shallow AW (green), and deep AW (orange) as simulated by the ECCOv4 in the Norske Trough for the period 1992–2015. The locations of the CTD stations are shown in Figure 2. $T > 0\,^\circ C$ and $S > 34.3\,\mathrm{psu}$ define AW.

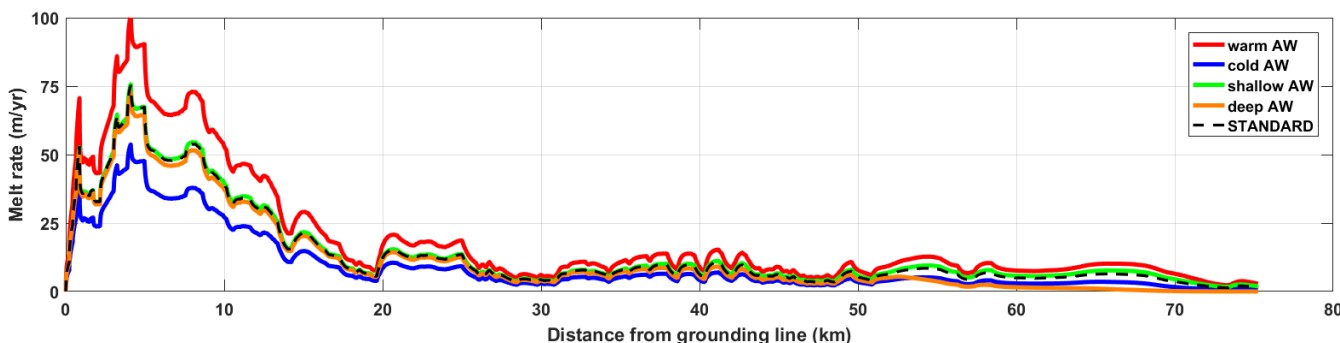

**Figure 7.** Submarine melt rates simulated by the 1D ISW plume model using variations in AW temperature and presence in the water column in the Norske Trough from the ECCOv4 for the period 1992–2015. The STANDARD CTD profile is the rift profile (black dashed). Warm AW (red), cold AW (blue), shallow AW (green), and deep AW (orange). The locations of the CTD stations are shown in Figure 2.





**Figure 8.** Sensitivity of the submarine melt rate along the centreline of the 79NG ice tongue due to (a) the entrainment coefficient, (b) the drag coefficient, and (c) the subglacial discharge. (d) shows the spatial variability of melting across the ice tongue and (e) the corresponding ice base.

(Figure 8a). Note that for the 200 % simulation a drag coefficient, $C_D$, of $8.6 \times 10^{-3}$ was used instead of the STANDARD $C_D$ = $9.7 \times 10^{-3}$, because for the STANDARD drag and an increased (200 %) entrainment coefficient the plume did not evolve. However, the influence of a reduced drag coefficient was found not to change the overall melt pattern (Figure 8b and Table



3). A smaller entrainment coefficient value yields lower melt rates (Figure 8a). The along-glacier melt rate distribution is not sensitive to variable entrainment coefficients.

**Drag Coefficient.** The STANDARD value for the drag coefficient $C_D$ used in this work is $9.7 \times 10^{-3}$ ($100\%$). To understand how the drag coefficient influences the submarine melt rates along the ice base, it is changed to $25\%$ and $200\%$ of the STANDARD value. Using a lower $C_D$, $25\%$ ($2.5 \times 10^{-3}$) and $50\%$ ($4.9 \times 10^{-3}$), results in a faster plume (not shown). In contrast, a higher drag coefficient ($200\%$; $1.9 \times 10^{-2}$) leads to a slower plume. Generally, the melt rates are less sensitive to the choice of the drag coefficient than to the entrainment coefficient (Figure 8a and b and Table 3).

A faster plume leads to higher entrainment of AW and, thus, more melting (Figure 8b and Table 3). In addition, the turbulent transfer coefficients for heat and salt, $\gamma_{T,S}$, depend linearly on the plume velocity (Eq. 7 and 8). As the plume accelerates, heat supplied by the AW is transferred faster toward the ice base. The drag coefficient only determines $\gamma_{T,S}$ as the square root and, hence, has a weaker influence. A slower plume causes lower mean melt rates following an increase of the drag coefficient to $200\%$ (Figure 8b and Table 3). The turbulent transfer of heat is reduced as a higher drag implies a decrease in velocity.

**Subglacial Discharge.** The subglacial discharge $D_{X0}U_{X0}$ is highly unknown for 79NG and here a wide range between $4.0 \times 10^{-4}$ and $7.5 \times 10^{-2} \, \mathrm{m^2 \, s^{-1}}$ was applied. The STANDARD value $4.0 \times 10^{-3} \, \mathrm{m^2 \, s^{-1}}$ is derived from the long–term (1958–2017) annual mean surface runoff on 79NG ($74.75 \, \mathrm{m^3 \, s^{-1}}$) from RACMO2.3p2 (Noël et al., 2018), where the subglacial discharge is estimated by distributing the surface runoff along the grounding line as a wide source. Decreasing or increasing the subglacial discharge by a factor 10 shows small changes in the melt rates along the centerline of the 79NG ice tongue (Figure 8c and Table 3), and is therefore assumed to be of little importance in this study. Ice shelf basal channels on the order of km's wide are found at Petermann Glacier in North Greenland (e.g., Rignot and Steffen, 2008; Dallaston et al., 2015). The effect of releasing the subglacial discharge through a concentrated, narrow opening of 1 km, yielding approximately $7.5 \times 10^{-2} \, \mathrm{m^2 \, s^{-1}}$, was therefore also tested. The simulations show an increase of roughly $13\%$ in both mean and maximum melting (Table 3).

### 3.5 Stability of the 79NG Ice Tongue

The simulated mean submarine melt rates, $m_{sub}$, are used to asses the stability of the 79NG ice tongue. A simple approach is followed that allows for an estimate of how long it takes for the ice tongue to completely melt down. The mean submarine melt rate along the centreline of the ice tongue is about $15 \, \mathrm{m \, yr^{-1}}$ (Table 4). The corresponding mean ice thickness, ice base plus ice elevation, is 256 m. The thickness within the GLZ is much greater. However, as melt rates are highest there it is reasonable to apply the mean thickness and the mean melt rate there as well. Taking into account the inflowing ice flux, $ice_{flux}$, and the mean surface melt rate, $m_{sfc}$, both provided by Wilson et al. (2017), the 79NG ice tongue would completely melt within 24 yr (Table 4). These estimates are of course sensitive to the chosen flow path (Figure 2) and associated melt rates and thicknesses (Table 4, south and north coast). It is also worth pointing out that the ice tongue thickens along the north coast due the very low melt rates compared to the inflowing ice flux. The variations in AW properties from ECCOv4 yield melt rates between $10 \, \mathrm{m \, yr^{-1}}$ (cold AW) and $21 \, \mathrm{m \, yr^{-1}}$ (warm AW) (Table 4). The corresponding melt times are 37 yr and 14 yr, respectively.



The melting furthermore leads to a freshwater flux (FWF) by the plume. The FWF can be quantified as:

$$FWF = \frac{(m_{sub} + m_{sfc} - ice_{flux}) \cdot L \cdot (\frac{2}{3} \cdot w_{up} + \frac{1}{3} \cdot w_{down})}{31557600 \cdot 10^6}, \tag{11}$$

where L is the length of the ice tongue, $w_{up} = 20\,\mathrm{km}$ is the width that is representative for the upper two-thirds of the ice tongue, and $w_{down} = 30\,\mathrm{km}$ is the width that is representative for the lower one third. The values in the denominator are used to convert from $\mathrm{km^3\,yr^{-1}}$ into Sv. The overall range in the FWF from both variable AW temperature and geometry is $11\,\mathrm{km^3\,yr^{-1}}$ (0.4 mSv) to $30\,\mathrm{km^3\,yr^{-1}}$ (1.0 mSv) (Table 4).

## 4 Discussion

Submarine melting is thought to play a key role in the mass balance and stability of the 79NG (Mayer et al., 2000; Thomsen et al., 1997). The associated freshwater fluxes are furthermore important to both local and large-scale ocean circulation (e.g., Bamber et al., 2018; Böning et al., 2016). It is therefore very important to constrain the submarine melt rates and to ascertain the sensitivity of the 79NG to environmental and model parameters. In the following we discuss the sensitivity of each of the parameters assessed in this study, and compare with previous findings.

### 4.1 Plume model melt rates

The results for the melt rates are qualitatively comparable with melt rates obtained from Mayer et al. (2000) and Wilson et al. (2017), but with some notable differences. Near the grounding line Wilson et al. (2017) calculated maximum values of 50 - $60\,\mathrm{m\,yr^{-1}}$, while downstream from 15 km along the rest of the ice tongue they found melt rates near zero. Mayer et al. (2000) found smaller melt rate, with a maximum of about $40\,\mathrm{m\,yr^{-1}}$ near the GL. Jenkins (2011) performed sensitivity experiments

| Model run | Mean submarine melt rate ($\mathrm{m\,yr^{-1}}$) | Mean ice thickness (m) | Mean melt time (yr) | Freshwater flux ($\mathrm{km^3\,yr^{-1}}$) / (mSv) |
|---|---|---|---|---|
| STANDARD | 15.2 | 255.5 | 24.2 | 19.7 / 0.6 |
| south coast | 11.6 | 218.5 | 29.5 | 13.8 / 0.4 |
| north coast | 1.3 | 201.4 | - | - / - |
| mean | 13.2 | 237.0 | 26.4 | 16.7 / 0.5 |
| warm AW | 20.5 | 223.2 | 13.7 | 30.4 / 1.0 |
| cold AW | 10.2 | 223.2 | 37.2 | 11.2 / 0.4 |

**Table 4.** Simulated mean submarine melt rates, calculated mean melt time of the 79NG ice tongue, and the freshwater flux that is released into the ocean due to melting for selected model runs.



with an analytical model and found an approximate linear relationship for mean melt rates near the GL region. This is consistent with the melt rate evolution identified here.

## 4.2 Sensitivity to ocean forcing

To investigate the response on the submarine melt rates and the plume dynamics below the 79NG ice tongue to a warming ocean, the temperature and depth of AW in the cavity were varied based on simulated variability between 1992 and 2015 in ECCOv4 (Figure 6 and 7, and Table 3). The submarine melt rates respond almost linearly to AW temperature changes (Figure 5); increasing the AW temperature leads to significantly higher melt rates (Figure 7 and Table 3). The identified linear relationship between ocean temperature and submarine melting is in agreement with Jenkins (2011). In contrast, Holland et al. (2008) found a quadratic relation for some Antarctic ice shelves and floating glaciers using a 3D ocean general circulation model.

This quadratic dependence is rooted in the heat transfer dependence of the melt rate across the ice-ocean interface, in which both ocean temperature and velocity plays a key role. In Holland et al. (2008), the flow is any velocity used in the parameterization for the melt rates, including the buoyancy driven circulation by meltwater. In the plume model used here, only the plume velocity U is considered in the heat transfer (Eq. 7) and, thus, in the parameterization of the melting (Eq. 5), with zero ocean (ambient water) speed (Jenkins, 1991, 2011; Smedsrud and Jenkins, 2004). This most likely causes the linear term of the fit to dominate over the quadratic term (Figure 5).

To be able to explain the plume dynamics in a warming ocean, especially entrainment and melting processes have to be considered (Jenkins, 1991). Entrainment of AW leads to a denser plume, in contrast, the increased freshwater input that results from melting acts to maintain the buoyancy of the plume. For the 1D ISW model, the plume velocity increases with rising AW temperature (not shown), suggesting that the freshwater input has a greater influence on the plume density than entrainment of AW. In addition, as the water temperature is continuously increased, more ice is melted whereas the entrainment rate remains constant. The enhanced freshwater amount accelerates the plume. Because the entrainment rate is parameterized with the plume velocity more entrainment of warm water is expected. This in turn leads to greater melt rates further downstream (Figure 7 and Table 3).

The range in AW temperature simulated by ECOOv4 is about 0.5 °C, which is similar to the warming observed by Schaffer et al. (2017). The accuracy of the ECCOv4 ocean state estimate depends on the model fields being well-constrained to actual observational data. On the continental shelf of northeast Greenland, in the area close to 79NG, the observational data coverage is sparse, which could imply that the AW properties in this region are not well constrained. However, AW flowing onto the shelf of northeast Greenland originates in the Norwegian Sea and Fram Strait, which are well-observed (e.g., Beszczynska-Møller et al. 2012; Onarheim et al. 2014). As a consequence, the range of AW temperature during recent decades in ECCOv4, and which is used here to force the 1D plume model, is consistent with the long-term warming of AW in the Fram Strait (Beszczynska-Møller et al. 2012) and in the Norske Trough (Schaffer et al. 2017). Simulated mean and maximum melt rates corresponding to an ocean warming higher than 0.5 °C (Figure 7 and Table 3) appear unrealistic (Wilson et al., 2017).



### 4.3 Sensitivity to model parameters

**Entrainment Coefficient.** The parameterization of the entrainment rate $\dot{e}$ used here was suggested by Pedersen (1980). It was tested during laboratory and field studies of dense bottom currents yielding reasonably results, especially for ice base slopes of 0.01 and less (Jenkins, 1991). However, the slopes within the GLZ along the centreline of the 79NG ice tongue are much

steeper, with a mean slope of about 0.03. This may thus lead to an overestimate in the melting. An entrainment coefficient value of $1.8 \times 10^{-2}$, half the value used by Jenkins (1991), was applied in the STANDARD run. We find this to be a reasonable constraint for the 79NG model configuration as the simulated melt rates are qualitatively comparable with results from Wilson et al. (2017).

**Drag Coefficient.** A higher drag coefficient $C_D$ slows the plume down and, thus, reduces the melting as less AW is entrained.

In contrast, a lower drag accelerates the flow, consequently leading to higher entrainment rates and, hence, elevated melt rates. The STANDARD value $9.7 \times 10^{-3}$ for $C_D$ was adopted from Jenkins et al. (2010) and assumed to be a good choice for studying ice-ocean interactions. Jenkins et al. (2010) derived this value mainly from observations of the hydrography and submarine melt rates calculated from radar measurements at the Ronne Ice Shelf.

The drag depends on the surface shape and the roughness. Nicholls et al. (2006) present ice base and hydrographic data

for the Fimbul Ice Shelf in Antarctica. One major finding is that the ice base of an ice shelf is probably much rougher than assumed before. Moreover, melt water channels that have walls as high as 50 m most likely cause a rougher surface and shape (Dutrieux et al., 2014). All this suggests a higher drag and we believe that such channels also exist at 79NG. Millgate et al. (2013) performed simulations with the MITgcm at Petermann Glacier and found a decrease in the mean melt rate associated with an increase in channel number. To take into account a rougher ice base and the effects of channels, we used the higher

drag coefficient suggested by Jenkins et al. (2010) as a suitable estimate for the 79NG ice tongue.

### 4.4 Spatial variability and subglacial discharge

**Spatial Variability.** A plume initiated at the GL rises and would flow along the Coriolis-favored side of the fjord (Millgate et al., 2013), which in case of 79NG is along the south coast (Figure 2). On the other hand, Wilson and Straneo (2015) found evidence of meltwater in Dijmphna Sund and suggest that the plume flows out at the northern terminus of the ice tongue and

into Dijmphna Sund. This is supported by Mayer et al. (2000) stating that the buoyant meltwater moves along the largest gradient at the underside of the ice base which is found to be the path along the north coast (Figures 2 and 8e).

The spatial variability of submarine melt rates across the 79NG ice tongue based on the three selected flow lines (Figure 8e) compares very well with results from Wilson et al. (2017). Along the northern side, melting is low except two peaks at approximately 25 km and 45 km where melt rates exceed 10 - 25 m yr$^{-1}$ (Figure 8d). This is due to the smoother ice base slope

compared to the centerline and the south coast (Figure 8e), as the plume velocity and entrainment of warm AW (Eq. 3) are proportional to the slope. This is consistent with results from Wilson et al. (2017) who found a significant correlation between melt rates and ice base slope at 79NG, with melt rates being highest where the ice base is deepest and the slope steepest. Maximum slopes within the GLZ along the centreline, the north and south coast are about 0.06, 0.13, and 0.03 (not shown).





According to Jenkins (2011), slopes of 0.01 are typical for the GLZ of ice shelves. The slopes for the centreline and the south coast are thus relatively steep and might explain the elevated maximum melt rates compared to results from Wilson et al. (2017), especially as this small band close to the GL is not included in the latter study.

**Subglacial Discharge.** The subglacial discharge enables the plume to evolve close to the GL. However, downstream convection is melt-driven (Jenkins, 2011) and controlled by additional freshwater input due to melting and, hence, not sensitive to variations in the subglacial discharge. The STANDARD subglacial discharge of $4.0 \times 10^{-3} \, \mathrm{m^2 \, s^{-1}}$ is the multi-year annual mean for 1958–2017 simulated by RACMO2.3p2 (Noël et al., 2018; Todd et al., 2018). A value of $4.0 \times 10^{-4} \, \mathrm{m^2 \, s^{-1}}$ is very low and might be applicable for a winter situation where no contribution form surface runoff to the subglacial discharge is assumed (Straneo et al., 2011). However, increasing or decreasing the subglacial discharge at 79NG by an order of magnitude leads to small changes in melt rate (Table 3). Particularly in winter, basal melting of the grounded ice stream might be a significant contribution to the subglacial discharge (Benn and Evans (2010), D. Slater, personal communication), which is not included in the simulated estimate from RACMO2.3p2 (Noël et al., 2018). Also the drainage or flow routing of the surface runoff will change over time, and is likely changing now (Karlsson and Dahl-Jensen, 2015), which adds to the overall uncertainty of the subglacial discharge. The STANDARD subglacial discharge of $4.0 \times 10^{-3} \, \mathrm{m^2 \, s^{-1}}$ is the best available value, but remains a basically unknown parameter.

Changing the distribution of the discharge influences the melt rates. Using a narrow, concentrated source leads to an increase of 13% in both mean and maximum melting. This implies that the spread along the grounding line is equally important as the actual total volume, so the effect of runoff and subglacial discharge may be greater if a more focused plume is assumed. To further investigate this question the number and sizes of cracks or channels need to be observed which is very difficult.

The RACMO2.3p2 surface runoff show an annual cycle and a postive trend (not shown), but it is not clear to what extent this is directly reflected in subglacial discharge. Other potential sources for subglacial discharge at 79NG are geothermal heat fluxes measured at NEGIS (Fahnestock et al., 2001), and a large marginal lake located north of the glacier (Thomsen et al., 1997). The drainage basin of 79NG is as large as $120\,000 \, \mathrm{km^2}$ (Seroussi et al., 2011) and thus accumulates forcing from precipitation (e.g., Cullather et al., 2016; Mernild et al., 2015; Ettema et al., 2009) and surface melting (e.g., Cullather et al., 2016) over this large area, further contributing to the uncertainty in subglacial discharge, but the contribution from geothermal heat fluxes appear small. Overall, the simulated melt rates presented here appear robust to variations in subglacial discharge (Figure 8c and Table 3) as model sensitivity is quite small to this quantity.

## 4.5 Stability of the 79NG Ice Tongue

The stability of the 79NG ice tongue was assessed by estimating how long it takes for the ice tongue to completely melt down in a warming ocean. The variations in AW temperature and presence in the water column drive melting between $10 \, \mathrm{m \, yr^{-1}}$ and $21 \, \mathrm{m \, yr^{-1}}$. In turn, this leads to a complete melt of the ice tongue within 37 years and 14 years, respectively, assuming a constant ice flux. These results clearly show that the ice tongue of 79NG is susceptible to thinning if the thermal forcing from the ocean is increased. Based on high-resolution satellite imagery, Wilson et al. (2017) argue that the ice tongue will retreat in the future and become ungrounded. They found a total melt flux of $14.2 \pm 1.6 \, \mathrm{km^3 \, yr^{-1}}$ exceeding the inflow ice flux of



$10.2 \pm 0.59\,\mathrm{km^3\,yr^{-1}}$. Further, Mayer et al. (2018) found that the 79NG ice tongue will most likely disappear within a few decades based on observations of surface features, ice thickness, and bedrock data.

For cold ($0.1\,^\circ$C) and warm ($1.4\,^\circ$C) ocean forcing scenarios, the freshwater flux ranges between $11\,\mathrm{km^3\,yr^{-1}}$ ($0.4\,\mathrm{mSv}$) and $30\,\mathrm{km^3\,yr^{-1}}$ ($1.0\,\mathrm{mSv}$). The warm AW case is similar to the $0.5\,^\circ$C warming found by Schaffer et al. (2017). The values obtained for the freshwater flux are on the same order as freshwater fluxes due to melting at other Greenland glaciers such as Helheim Glacier, Petermann Glacier ($11.7 \pm 1.4\,\mathrm{km^3\,yr^{-1}}$) (Wilson et al., 2017), Jakobshaven Isbræ, and Kanquersal Glacier.

## 5   Conclusions

In this study submarine melt rates of the 79NG have been examined based on new observations of hydrography and a 1D Ice Shelf Water plume model. The observations show that warm Atlantic Water (AW) with temperatures up to $1.0\,^\circ$C is present at grounding line depth. The AW thus has the potential to drive submarine melting along the ice base, confirmed by observed mixtures of AW and meltwater in the cavity. To improve our understanding of ice-ocean processes, as well as the sensitivity to ocean warming, the sensitivity towards a number of parameters has been tested.

The melt rates for the plume model are comparable with, yet higher than, results from Mayer et al. (2000) and Wilson et al. (2017) who used ice flux divergence calculations. The simulated melt pattern is qualitatively similar. Maximum melting up to $76\,\mathrm{m\,yr^{-1}}$ is simulated near the grounding line where the melting is hard to estimate based on remote sensing, so this might explain the overall higher mean melt rates. The resulting freshwater flux to the continental shelf ranges between $11\,\mathrm{km^3\,yr^{-1}}$ ($0.4\,\mathrm{mSv}$) and $30\,\mathrm{km^3\,yr^{-1}}$ ($1.0\,\mathrm{mSv}$). This is between 5 - 12 % of the total freshwater flux from the Greenland Ice Sheet, $3\,200 \pm 358\,\mathrm{km^3}$ since 1995 (Bamber et al., 2012).

A clear influence of AW temperature is found on the simulated 79NG melt rates, and AW temperature variability of about $\pm 0.5\,^\circ$C is documented here from the new ECCOv4 ocean state estimate for the 1992–2015 period. Corresponding to this range, the 1D Ice Shelf Water plume model simulates mean melt rates between 10 - $21\,\mathrm{m\,yr^{-1}}$. Using the 1D Ice Shelf Water plume model a mean melt rate of $15\,\mathrm{m\,yr^{-1}}$ is simulated and maximum melting between 50 - $76\,\mathrm{m\,yr^{-1}}$ . Melt rates increase linearly with AW temperature, consistent with Jenkins (2011). A long-term general warming of the AW since the 1970's of about $1.0\,^\circ$C (Onarheim et al., 2014) would thus imply that mean melt rate of 79NG has doubled since then. The melt rates are not sensitive to change in vertical extension of AW over a large $\pm 100\,\mathrm{m}$ range. This is because the major part of the melting takes place near the grounding line, and little at the upper depth of the AW. Melt rates are also not sensitive to a an order of magnitude change in subglacial discharge at the grounding line.

The 1D Ice Shelf Water plume model has a depth-integrated mixed layer, no across flow variations, and assumes steady state. These assumptions should be kept in mind when interpreting the results, but given the very limited set of observations the model complexity appears sound. Moreover, even though the 1D ISW plume model is based on simplified physics the melt rate pattern and magnitude are comparable to other studies (e.g., Wilson et al., 2017; Mayer et al., 2000). In addition, the sensitivity studies performed here help to better understand submarine melting and plume dynamics of more sophisticated models.





New observations of hydrography and improved topographic mapping of the 79NG will further improve our understanding of melting in this area, and allow for better estimates of the sensitivity of the ice tongue to future warming. More extensive observations would also allow for an evaluation of how representative the CTD profile used here is for the full length of the 79NG, and to asses many of the results presented here.

*Code and data availability.*  Model code, as well as ice shelf bathymetry, and ocean forcing of the STANDARD case will be made available at https://doi.pangaea.de. A Matlab file to plot the basic Ice Shelf Water plume properties is also made available. We are currently working to get the relevant DOI number. For the review process data can be downloaded using the following link (Anhaus et al., 2019):

https://www.pangaea.de/tok/5b00694d7518bfbfdae8f4e6e10551c4314743bf

*Author contributions.*  The study was designed by Lars H. Smedsrud, Marius Årthun, and Philipp Anhaus. Data acquisition (hydrography)
were performed by Fiammetta Straneo. Philipp Anhaus performed the model runs, analysed the data, and created the figures with significant input from Lars H. Smedsrud and Marius Årthun. Lars H. Smedsrud provided the 1D ISW plume model and technical support. Marius Årthun provided the data for ECCOv4 and technical support. Interpretation of the results were performed by Philipp Anhaus, Lars H. Smedsrud, Marius Årthun, and Fiammetta Straneo. Drafting the first version of the manuscript was done by Philipp Anhaus based on his Master's Thesis "Circulation and Ocean Driven Glacial Melting in a Greenland Fjord" completed at the University of Bergen in December 2017. Lars
H. Smedsrud, Marius Årthun, and Fiammetta Straneo provided critical revisions and important additions during all stages of manuscript preparation. All authors give final approval for the publication of this manuscript in its current form.

*Competing interests.*  The authors declare that they have no conflict of interest.

*Acknowledgements.*  This research was funded by the Fast Track Initiative 2018 from the Bjerknes Centre for Climate Research, Bergen, Norway and the Geophysical Institute of the University of Bergen, Norway. Marius Årthun was funded by the Research Council of Norway
project PATHWAY (Grant 263223). Lars H. Smedsrud was supported by the Ice2Ice project (ERC Grant 610055) from the European Union FP7/2007-2013. We thank Donald Slater from the Scripps Institution of Oceanography, La Jolla, California, USA for providing the subglacial discharge estimates dervied from the RACMO2.3p2.



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
