# Peer review of "Sensitivity of submarine melting on North East Greenland towards ocean forcing"

_The Cryosphere, 2019_

## Referee Comment (RC1) · Anonymous Referee #1 · 2 May 2019

Mass loss from the Greenland Ice Sheet is important to sea-level rise and ocean freshening. The rate of ice loss from some parts of Greenland is modulated by interactions with ocean melting. This study reports the results of a one-dimensional plume model of a layer of meltwater beneath the floating tongue of 79N glacier in north-west Greenland.

The model is first applied in a 'standard' case, with the goal of explaining details of the melting pattern beneath the ice tongue, and then several perturbations to the model are applied. Notably, the temperature of the ocean waters forcing the plume model is varied, and the response is taken as a study of the sensitivity of 79N to ocean warming.

This 1D plume model is a highly simplified representation of ocean flow beneath floating ice, and is not a 'state of the art' model, as full 3D ocean models are now commonplace. Therefore, in my opinion the reader needs to be convinced that this simple model can capture the relevant physics, before they can believe the sensitivity of melting to ocean temperatures that emerges from this model. As detailed below, I am not sure that the present paper demonstrates the utility of this simple model.

Broader points

The model neglects almost all of the complexity of flow beneath floating ice, including Coriolis, tides, 3D flows in complex topography, shelf-driven circulation, flow around the island, etc. Therefore, this model needs to demonstrably reproduce observations in order to be credible. I did not find that the current manuscript demonstrates this. On several occasions the paper cites the remote sensing melt rates of Wilson et al, but I didn't see an explicit comparison of the plume model results to those observations. In the absence of that comparision, the melt rates appear to be significantly too high. On Page 14 the authors quote a balance melt rate of 4.2 m/y. The 'standard' simulation has a melt rate of 15.2 m/y. Is 79N glacier thinning at a rate of 11 m/y? Such a thinning rate should be easily visible from satellites. Assuming instead that the ice shelf is in balance, I infer the plume model melting is too high by a factor of 3.5. This suggests that the plume model melting sensitivity to warming is also much too high, and this weakens the credibility of the study. The authors could address this point by explicitly validating the plume model melt rates against observations, using satellite-derived melt rates that take into account any thinning in the ice shelf. They should validate melt rates along the plume path, and also in a mean ice shelf sense.

The apparently high melt rate does not really decrease in any of the sensitivity studies in table 3, apart from the one in which the entrainment coefficient is decreased further. But even in the standard simulation the entrainment coefficient is already at a very low value, relative to the literature, and the perturbed value is a full order of magnitude lower than the value recommended by Bo Pedersen. Thus it may appear that the plume model is structurally incapable of reproducing observed melt rates, as a result of its simplified physics.

I think the whole CTD cast is being specified as the 'ambient' water for the plume (page 10). However, this is circular, since the upper part of the CTD cast already contains the meltwater that is the 'plume', as evidenced by Figure 3c. In other words, the 'answer' is being specified in the 'forcing'. It would be a more valid experiment to specify the pure source water, i.e. the warmest densest AW only at the bottom of the CTD, and then see if the plume model can generate the observed colder meltwater in the upper part of the CTD. Since this approach would warm the ambient waters relative to those used in the experiments, I infer that it would even further exacerbate the excessive melt rates.

The authors discuss whether their melting sensitivity to ocean temperature is linear or nonlinear. I have several comments: i) The authors report a quadratic fit in Figure 5 but seemingly only based upon the 7 warmest temperatures. Why not use all of the temperatures? ii) They later claim that the fit is linear for the 7 warmest temperatures, which is fine, but that linear relation cannot be universally true since it does not pass through zero melting for zero thermal driving. So the nonlinear fit must be the more general relationship. iii) The linearity or otherwise is not rigorously tested using a statistical test. iv) On page 20 some reasons for the linearity are stated. As described above, I think the results are entirely consistent with the quadratic fits of Holland et al 2008 over the wider temperature range, and so there is no discrepancy to explain. Further, any discrepancy that is present would most obviously be explained by the lack of Coriolis force in the plume model.

The stability of the ice shelf is discussed on page 18 and Table 4 as if it is a passive ice body that simply melts away in response to a perturbed ocean melt rate. There are several problems with this: i) Ice thinning will induce ice feedbacks, such as enhanced discharge, which could stabilise the ice. ii) The ice shelf would collapse long before it melted to zero thickness. iii) Ice thinning will induce ocean feedbacks, such as decreased melting as the ice thins into colder waters.

More specific points

General: No attention is paid to seasonality of the subglacial input?

General: are tides important?

Title: suggest changing to 'Sensitivity of submarine melting of 79N glacier to ocean forcing'?

Abstract and elsewhere: there is a claim of 5% and 12% of total Greenland freshwater flux. What does this mean? Is this claiming a fraction of the steady state ice discharge from Greenland, or a fraction of the unsteady mass imbalance of Greenland, or perhaps a fraction of the total Greenlandic ice melted by the ocean, or its unsteady component?

P2 L10: clarify the extent to which the preceding discussion was relevant to 79N glacier. I think it was mainly about the fjords to the south, which do not have ice tongues. Is 79N losing mass?

P2 L26: The papers cited are not primarily observational.

P2 L29: Distinguish between meltwater and glacial modified water?

P3 L8 and other places: Ice Shelf Water (ISW) is a recognised water mass, meaning water below the surface freezing point. There is very little ISW here, so re-name this to a meltwater plume model or similar.

P6 L11: Melt rates are high due to high slope, not pressure depression of $T_b$, which is small.

P7 L12 onwards: This paragraph is very confusing. I couldn't follow most of the sentences in it.

P9 L6: With the 'veering', are the authors referring to the very slight deflection of the line within the inset of figure 3c, seemingly from one meltwater mixing line to another? I couldn't follow why that is necessarily caused by runoff. It could be caused by mixing between water masses, or two sources of AW driving melting in different locations?

P10 L5: need to define the upper limit of AW

P10: figure 8 is cited out of order.

P13 L16: what does 'in a 2D or 3D concept' mean?

P13 L34: no iceberg calving?

P17 L2: The plume did not evolve? why not?

P20 L20: the entrainment rate is not constant.

---

## Short Comment (SC1) · 9 May 2019

Thank you very much for investing time and providing constructive feedback, corrections and useful comments. You rise a number of important points and provide several helpful suggestions.

We will indeed revise the manuscript accordingly and make appropriate changes to improve this work. In the revised version we will seek to further compare our melt rates explicitly with other studies and improve the explanations about the physically based, but fairly simple 1D meltwater plume model used here.

Sincerely, Philipp Anhaus

---

## Referee Comment (RC2) · Anonymous Referee #2 · 21 May 2019

Summary:

In my view this manuscript presents an application of an existing 1D plume model to the 79NG geometry. The model is constrained by a single CTD profile collected during one summer (described by Wilson et al). Elsewhere the model uses standard values of unknown variables. This CTD profile is then modified in four different ways to simulate AW temperature and/or thickness change, and the melt rate is recomputed from the plume model for these cases. The resulting melt rate for each case is then used in combination with an ice flux value estimate (satellite derived, Wilson et al) to give a time over which the ice tongue will melt under the prescribed AW, assuming nothing but the AW vertical profile changes.

Major comments:

[Figure]

1) The ice-tongue melt-time projection neglects all feedbacks in this glacier fjord, including ice dynamics (evolving ice flux across the grounding line, and shape of the cavity), iceberg calving, evolving ocean circulation and wind patterns (and sea ice, if relevant) outside the fjord. The projection also does not take into account, rotational effects, dimensionality of the problem, and temporal changes inside the fjord - I would expect at least seasonal AW temperature/thickness (and presence), and seasonally distributed subglacial discharge to play some role here. As a result, I am not sure the provided ice-tongue melt-time projection has much meaning. If not a coupled glacier-ocean model, I think at least a glacier model should be involved, to produce a somewhat more robust statement about the 79NG stability at present and in the future.

2) The main focus of the manuscript is on assessing sensitivity of the plume model to its parameters - but I am not sure it provides any new insights or conclusions. I am also not sure it provides any new insights on the processes driving submarine melting - but is it possible that I missed it - it would be helpful if the authors clarified what the contributions are and how they differ from previous studies. Here are a few studies (not referenced here) that have done this before and more exhaustively: Carroll et al 2015, and 2016, Sciascia 2013, Beckmann et al 2018. As it stands, I think the presented plume model application to 79NG is within the parameter range studied previously. If not, it would be good to clarify that. The main result (linear scaling of melt rate with AW temperature) is consistent with other studies in Greenland glaciers, and as mentioned even in the discussion here it maybe more of a property of the plume model itself, than anything else.

3) The manuscript is not very carefully referenced. Although there are a lot of references, the choices are sometimes quite arbitrary. Given this is primarily a sensitivity study of the plume model - it should be clear how the findings here differ from other (often more complete and insightful) sensitivity studies of the plume model. Modeling studies are at times used as references where one would expect a reference for observations.

4) I feel the manuscript is written quite confusingly and could use a bit of reorganization. Description and interpretation are often mixed without a clear distinction. Although I have quite a few in-line comments and clarification suggestions below, I feel that for what the manuscript does, it could be half its length, and more to the point. A lot of the confusion arises because of the poor organization. Here are some suggestions for restructuring:

*Include a background section (could be part of introduction) - where all relevant information about the region and glacier is summarized and refer to it whenever necessary, in stead of giving background throughout the manuscript, sometimes repetitively, sometimes not at all.

*Clearly describe the experiment setup in the methods section, motivate and justify these experiment choices and clarify what you are trying to achieve. State all the assumptions in this section. At present, while there is discussion of sensitivity to some parameters in the end, there are many model choices for which there is no reasoning/explanation provided

*Describe the relevant part of the results (I think currently the result description is quite long, given it doesn't provide that much new insight)

*Explain why the results (resulting melt rates in this case) are believable for the base case (present), and only then move on to the results for the future case.

*To discuss the future warming scenarios, I think some sort of a model would be needed (see points 1)

In-line comments:

Abstract:

L7 - decay of what? 20 km from the grounding line? L10 - why is the melt rate sensitivity reported along a centerline when just a few lines above the melt rate is divided into three sections? Also, is this range of melt rate or melt rate increase? L13 - In

which way does the manuscript improve the understanding of processes driving submarine melting of marine-terminating glaciers around Greenland? This is the place to be specific.

Intro:

P2: L5 - general statement - doesn't need a reference. L6 - Holland 2007 seems a more appropriate reference here, than this review paper. L7 - Do you mean net mass loss increase? Also, this sentence seems to contradict a previous sentence (P1L20-22: The enhanced mass loss is caused by increased surface melt, and retreat and speed up of marine-terminating glaciers (Enderlin et al., 2014)) L8 - The increase of submarine melting.... not the submarine melting itself - ...leads to an inward migration..... A stable glacier can still have submarine melt. L9 - "It is important to study submarine melting since it is a likely trigger of change of ice loss from the ice sheet." Again, the presence of submarine melting is not a trigger of change, it is the the change of submarine melt that may act as a trigger of change. L14 - buttressing is defined usually at the grounding line, not at the terminus, could you specify what you mean by buttressing at the terminus here? L26 - Are these references supposed to refer to observations of melt/discharge driven plumes? Two of these are models, not observations L27 - which of the two scenarios are likely to happen at NG? and why is it likely? The majority of the subglacial discharge is most likely released at depth —- are you referring to 79NG specifically here? again, why is it likely? is there any support for this, or is that an assumption (which is completely fine as long as it is clarified) based on observations elsewhere? L29 - This is another awkward choice of reference. There have been plenty of earlier studies characterizing channelized network under ice tongues/ ice shelves. Since Dallaston does not relate to 79NG or even Greenland specifically, I don't understand the choice for this particular reference here as opposed to earlier ones. L33 - Isn't it 50% calving and 50% submarine melt? The wording here suggests that 50% is from these two together and the remainder from something else. L34 - This manuscript really overuses the word likely. It would help to clarify what is known (reference), what

is speculated (reference), and what is assumed for the purposes of this study.

P3: L5 - In my view this is more of a sensitivity study, not a process study

Data: P3: L20 - What is a high-res digital bathymetry model and how does it compensate poor data coverage? L18-23 - Bathymetry - did you do this data merging here for the purpose of this manuscript? If so this needs to be described in much more detail. If not, it would be good to first write what product is used in which part of the MS, followed by a brief description what this product consists of.

L24 - This third paragraph logically follows from the first one, not from the second one as it refers back to the ice, so perhaps rearrange. Also please explain the choice of the spatial filter, and how it guarantees a increase of the ice base, or if this was then enforced by some other procedure. L25 - How were the plume paths chosen? If it is discussed later perhaps reference the section.

P4: L1 - Which tidal effects? Do you include those here or you use the basic version? L5 - Why is STANDARD capitalized? At least at this point of the manuscript this is not at all clear. L5-7 - This part is a bit rushed, could you be a bit clearer on how you derive the subglacial discharge, and what assumptions go into the derivation. Further, have you considered separating the summer and winter case? Presumably the subglacial discharge is very seasonal and unless the plume model depends linearly on this parameter, using a long term annual mean might over- or under-estimate the melt. L9 - I am not sure I am familiar with the terminology "line source equation" could you clarify what this term means? L9-10 - it is the quantities, not the fluxes of the quantities that are conserved.

P5: How is the "re-circulation" and "the southern and northern recirculation in Fram Strait" shown in the figure?

P6: L7 - What value do you use for the ref. density? I don't see it in the table. L8 - Clarify that initial doesn't refer to initial condition as there is no time dependency in the

equations. L9 - Why is T0 set to freezing point? Does the result depend on different values of T0? What is S0 set to? L11 - Melt rates at 79NG are .... This statement is not specific to NG, or is it? L24 - Do you have any reference that -15C is reasonable, or how sensitive is the result to that?

General - I think it would be clearer if the plume model was first presented in general, and only after all the concepts are introduced, you can introduce specific choices for 79NG and justify how appropriate they are. Constantly switching between these tow makes it very confusing.

P7: Table 1 - the values used for the constant, where are they taken from? - reference L6 - Having determined that rotation is important in this fjord, how is it taken into account? As far as I am aware applying 1D plume model to 3 different paths, does not deal with rotation effects - but that is what line P7L9-11 seem to suggest. L8 - is 2-layer an assumption, or is it an approximation based on observations? L9 - define f when it is introduced not several lines later L11 - The differences between.... it seems like this belongs more to discussion/results than here. L13 - Since there is no dependency, aren't all variables diagnostic?

P8: L4 - I don't understand this what ....with S instead of Sb refers to L5 - can you show that ECCOv4 does a reasonable job in this region? are there any data to constrain it here, if so, how well does it match them, if not, why do you think this coarsely resolved model represents well the water masses relevant to your computation? L7 - Forget et al., 2015 - Another incorrect reference. Also, this reference is not even listed under the "References" section L13 - Again, I don't follow why it is reasonable to consider a long-term mean for the value of subglacial discharge. At a Greenland glacier like NG79, subglacial discharge will vary seasonally, if it has a significant surface runoff component. I think that would only make sense if the basal melt scales linearly over the range from 0 subglacial discharge (winter) to max subglacial discharge (summer) - is that the case at NG79? L14 - Could you please give more detail on the surface run-off calculation. L15 - Could you please provide reasoning for the assumption of

equally distributing the flux?

Results: P8: The first part of results is more of an introduction - the data and water-masses discussed here are presented elsewhere, so these aren't quite "results" of this paper. P9: Did you use just one horizontal grid cell profile from ECCO or some spatial average (which would be more robust)? Why did you choose this particular grid cell? It does not seem to be so well justified, because water there is saltier than in the observed CTD profile (as shown in Fig. 3c). Why is the chosen ECCO grid cell so far from the observations, how coarse is the model resolution here (km) - can you show ECCO bathymetry for comparison? P10: L5 - what do you mean by "not significant in mean"? L7 and elsewhere - clarify when you are talking about observed and when about modeled AW. L8 - Is the AW protected from heat loss in the winter as well? Does sea ice form in this area, and is it possible that it transforms AW seasonally, in the winter - something that the model with this resolution might not resolve well? L10 - why three flow lines? L15 - introduce figures in their numbered order L22 - do you mean retreat? General - How does the thickness of AW change through time? Is there any evidence of AW in the cavity all year round? Is it possible that AW is present in the cavity only in the summer (e.g. due to heaving isopycnals?), what controls the renewal/circulation on this fjord L18 - Optimal in which sense? how did you determine that?

P11: L2 - Motivation/justification of model parameters should be in the methods, or early on, not in the end after the results have been described. L2 - result not results L5 - 10 km....looks even within first 2 km to me L6 - what is the definition of GLZ here? Fig 4, are the jumps in velocity and thickness simply a function of the ice base profile or its derivative - can you add the relevant quantity to the plot here? I see now it is plotted in Fig. 8, but here it would be appropriate as well since this is the first case discussed.

P12: L1 - gradually approaching -2.0 C (Fig 4d shows the limiting T to be more like -1.7 or -1.8 C) L2 - In contrast is the density contrast (reword) L7 - what is the reason for the sudden increase at $\sim$72 km? L8 - reference an equation that implies more AW entrainment with higher velocity, since the amount of entrained AW is not plotted. Same

for next sentence. L12 - ....the density contrast continuously decreases as a result of less melting - Fig. 4a doesn't quite show a decrease of melt part 30 km, in fact the melt rate is more less constant past this distance.

P13: L8 - accelerates linearly do you mean that the acceleration is constant over the first 5 km? There are still jumps in the velocity L9 - Besides three local minima - which ones? I see lots of (>3) local minima on the quite rugged velocity plot L11 - What is the reason for the velocity decrease there? L11 - "Small scale variability is due to the ice base." Clarify that point earlier, when discussing melt rate already - since the small scale features are already visible there. L17 - a 2D or 3D concept .... the units suggest that you are extrapolating the result to 3D L18-21 - first introduce/justify the experiments, then describe them L25 - At the end melting variability - at the end of what? L31 - why 0.5C? L34 - ... and outflowing flux down-glacier? Or do you just mean flux across the grounding line?

P14: L1 - the ice flux is probably quite spatially variable (3D) and possibly also varies seasonally. Using a bulk value is not very well justified. Alternatively, a use of glacier model would be more appropriate to assess the ice tongue stability. L2 - in a submarine melting of about $4.2myr-1$ - 4.3 no? This whole section is very confusing and involves a lot of hypothesizing, and if anywhere, it probably belongs to discussion, not to results (that is the description of the outcome of the experiments) I missed where equations 9 and 10 come from, what they mean, and what they assume

P15: L5 - because ECCOv4 does not have an ice tongue - this is not a justification for your assumptions. As mentioned above - is there any evidence that AW is in the cavity year-round? Section 3.4: Can you comment on if distributed plumes would be a better model here or not?

P17-18: I think the sensitivity of the plume model to various parameters have been addressed more exhaustively in the past (see general comments). How do the selected experiments here extend/complement/contradict the sensitivity findings of the previous

studies?

P18: L24 - No calving? Ice dynamics? L32 - Is the evolving thickness of the ice tongue taken into account in the mean melt time calculation? (it should be) I am unconvinced that assessing future glacier stability simply based on a far-field temperature and a fixed glacier flow-line geometry is of much relevance/value. If this number should have any meaning, there should be some evidence/justification, that the more complicated processes are well captured by the simple plume model.

P19: L14 - what do you mean by "qualitatively"? L15 - why do you think there are such large differences in the GL melt rates? Table 4 - do you use mean melt

P20: L1 - "approximate linear relationship for mean melt rates near the GL region." - melt rate linear with what other quantity? L25 - typo, ECCOv4 L30 - I am not sure the use of ECCO is really adding much to the manuscript - since it is basically only used to get the past temperature variability, and the only justification that it does well, is that the range of variability is similar to the range of variability in the Norwegian Sea and Fram Strait. Why don't you just use the temperature range the observations show? If you use ECCO, you have to go on to justify it does well in this region. L32 - why are they unrealistic? Could you please explain?

P21: L27 - spatial variability - do you mean across-flow or along-flow? L31 - how accurate is the satellite-derived melt rate near steep ice basal topography when the ice is not expected to be in hydrostatic equilibrium?

P22: L21 - why not? where else would the meltwater go? L31 - "assuming a constant ice flux" not a justified assumption Is any of this thinning that is apparently already underway observed in the rate you/Wilson et al suggest?

P23: L6 - reference for the other 3 glaciers? Where is Kanquersal Glacier? L8 - existing observations, not new - as far as I understand there were no new observations/data presented in this paper L11 - there would be an AW/melt mixture even if the melting
was driven by subglacial discharge, would it not? L11-12 awkward sentence L13 - where are these differences and what are they attributed to? L18 - what is since 1995? I though you are talking about annual freshwater flux.

---

## Short Comment (SC2) · 4 Jun 2019

Thank you very much for investing time and providing very detailed and constructive feedback, corrections and useful comments regarding our paper.

We will start improving this manuscript by thoroughly re-organizing and re-referencing. We will focus on pointing out more clearly new insights on submarine melting at the 79NG. The approach to assess the stability of the 79NG is indeed based on weak assumptions and neglects feedbacks in the fjord system. We seek to reform this by at least including seasonal AW temperature/thickness (and presence) and seasonally distributed subglacial discharge. We will further intensively discuss the suggestion to include a glacier model.

[Figure]

Sincerely, Philipp Anhaus

---

## Author Comment (AC1) · 9 Jul 2019

**Dear Anonymous Reviewer #1,**

Thank you for your review and your constructive suggestions. We have worked hard to improve the paper, and hope that you will be satisfied with our response. Below we have responded to all of your suggestions using bold font, and most of your suggestions have been applied in the resubmitted version of the paper marked in red.

**Best regards,**

Philipp Anhaus, Lars H. Smedsrud, Marius Årthun, and Fiammetta Straneo

**Broader points**

1) Model neglects complexity, Comparison between simulated melt rates using the 1D meltwater plume model and satellite-derived melt rates from Wilson et al., 2017

The model neglects almost all of the complexity of flow beneath floating ice, including Coriolis, tides, 3D flows in complex topography, shelf-driven circulation, flow around the island, etc. Therefore, this model needs to demonstrably reproduce observations in order to be credible. I did not find that the current manuscript demonstrates this. On several occasions, the paper cites the remote sensing melt rates of Wilson et al, but I didn't see an explicit comparison of the plume model results to those observations. In the absence of that comparison, the melt rates appear to be significantly too high. On Page 14 the authors quote a balance melt rate of 4.2 m/y. The 'standard' simulation has a melt rate of 15.2 m/y. Is 79N glacier thinning at a rate of 11 m/y? Such a thinning rate should be easily visible from satellites. Assuming instead that the ice shelf is in balance, I infer the plume model melting is too high by a factor of 3.5. This suggests that the plume model melting sensitivity to warming is also much too high, and this weakens the credibility of the study. The authors could address this point by explicitly validating the plume model melt rates against observations, using satellite-derived melt rates that take into account any thinning in the ice shelf. They should validate melt rates along the plume path, and also in a mean ice shelf sense.

These are important concerns and suggestions, and we agree that this should be improved in the paper. We will thus implement a better comparison by plotting the satellite-derived melt rates on top of the simulations from the 1D ISW plume model (Figure 4a). However, the satellite-derived melt rates are also estimates and have their own assumptions, and cannot be regarded as "truth". The plume model uses high resolution topography and observed ocean temperatures, so it is also based on observations. In the work from Wilson et al., 2017 a small band downstream of the grounding line is excluded, because it is hard to measure that part from space (Wilson et al., 2017 and Wilson N. (2018), personal communication) and they do not show submarine melt rates from this area, where highest melt rates are expected and simulated by the plume model. Maximum melt rates found by Wilson et al., 2017 are between 50 m/yr and 60 m/yr downstream from the neglected band. We found, that this coincides with the results from the plume model.

One assumption in Wilson et al., 2017 is that the floating ice tongue of the 79NG is in hydrostatic equilibrium with a constant ice density of 920 kg/m3. They state that hydrostasy is a good approximation over sufficiently long horizontal length scales and shallow tongue thickness gradients based on the work by Brunt et al., 2010. However, near the grounding line the tongue thickness gradients for the centreline and the south coast are large (Figure 8e in our manuscript; Mayer et al., 2000; Schaffer, 2017; Mayer et al., 2018) and the slope is steep (Figure S1). Within that area the hydrostatic assumption is less justified and, thus, Wilson et al., 2017 excluded data within a few

kilometres of the grounding line. Downstream of 5.8 km from the grounding line, simulated melt rates from the plume model are 50 m/yr and less (Figure 4a in our manuscript) and, thus, in our view, comparable to results from Wilson et al., 2017.

Figure S1: The slope of the ice base sin¢ of the 79NG with respect to the distance from the grounding line in km along three flow lines from RTopo2 (Schaffer et al., 2016). Centreline (black), South Coast (blue), and North Coast (red). The flow lines are marked red in Figure 2b in our manuscript.

In their Figure 2 (lower), Wilson et al., 2017 show the melt rates in a transverse profile of the 79NG. There are clearly two peaks visible with melt rates of about 75 m/yr, supporting the results of the plume model. The transverse profile is taken approx. 1 km downstream of their defined upstream flux gate.

We have additionally actually used the satellite-derived melt rates from Wilson et al., 2017 to constrain some of the model parameters. This led to the choice of the entrainment coefficient to be 0.018. The overall important coefficients (entrainment and drag) have a large range in nature, and sub-ice-shelf values are uncertain, we therefore made sure that the simulated melt rates are in reasonable agreement with the satellite-derived melt rates from Wilson et al., 2017.

The plume model is well tested (Jenkins, 1991; Smedsrud and Jenkins, 2004; Jenkins 2011; Schaffer, 2017; Mayer et al., 2018) - and captures the important physics. Mayer et al. (2018) recently used the plume model to simulate thinning rates of the Midgardsormen, a part of the 79NG ice tongue. We have also tested the sensitivity to tidal flow and found that the tides are too small to affect the melt rates (Anhaus, 2017, Master's thesis, unpublished). The new text about tides will be found in Section 2.4.

Reeh et al., 1999 estimated the total freshwater volume from the 79NG produced by submarine melting at about 13 km3/yr. This compares well to the freshwater volume simulated by the plume model of 19.7 km3/yr (STANDARD, Table 4), and we are thus confident that the plume model does a good job.

2) Entrainment coefficient

The apparently high melt rate does not really decrease in any of the sensitivity studies in table 3, apart from the one in which the entrainment coefficient is decreased further. But even in the standard simulation the entrainment coefficient is already at a very low value, relative to the literature, and the perturbed value is a full order of magnitude lower than the value recommended by Bo Pedersen. Thus it may appear that the plume model is structurally incapable of reproducing observed melt rates, as a result of its simplified physics.

It is indeed correct that the melting depends on the entrainment coefficient. But by following the observed topography at horizontal resolution, and constantly calculating plume speed and thickness, this plume model should be better capable of simulating realistic melting than a coarse 3D model. To learn how sensitive the 79NG is to increased ocean temperatures an ocean model is clearly needed. All models are wrong – some are useful. So, we hold that this 1D plume model is

useful, and give new insight based on the simple, but sound, assumptions of a constant entrainment coefficient. The suggestion by Pedersen (1980) yielded reasonable results for ice base slopes of 0.01 and less. However, the slopes within the GLZ along the centreline of the 79NG are much steeper, with a mean of about 0.03 and a maximum of 0.06 (Figure S1).

3) CTD

I think the whole CTD cast is being specified as the 'ambient' water for the plume (page 10). However, this is circular, since the upper part of the CTD cast already contains the meltwater that is the 'plume', as evidenced by Figure 3c. In other words, the 'answer' is being specified in the 'forcing'. It would be a more valid experiment to specify the pure source water, i.e. the warmest densest AW only at the bottom of the CTD, and then see if the plume model can generate the observed colder meltwater in the upper part of the CTD. Since this approach would warm the ambient waters relative to those used in the experiments, I infer that it would even further exacerbate the excessive melt rates.

It is correct that the observed CTD profile contains a fresh plume, but only in the upper 100 m. Usually the plume detaches at this depth (Figure 8e), but this is also approximately the depth of the ice front. This will be explained more clearly in the text.

Additionally, to investigate this further we performed a model simulation containing only the warmest, densest AW in the CTD profile obtained in the rift of the 79NG in September 2009. This profile contains 271 entries for depth (-600 m to -60 m), temperature (0.99 °C), and salinity (34.66 psu). We found only minor changes. The mean melt rate increased from 15 m/yr in the STANDARD case to 17 m/yr. The maximum melt rate increased from 76 m/yr to 80 m/yr.

However, the plume path increased from 75 km to 80 km in response to the saltier ambient water close to the ice base. A greater buoyancy difference was thus simulated between the ambient and plume water (Figure S2), and it takes longer for the plume to get neutrally buoyant. In this case the plume is not neutrally buoyant at 80 km, this is only the length of the ice tongue. As shown in Figure S2 is the density contrast above zero.

| 0.2 Den |        |  |                                                                                                                  |   |    |
|---------|--------|--|------------------------------------------------------------------------------------------------------------------|---|----|
| C 000   |        |  |                                                                                                                  |   |    |
| Li 10.4 |        |  |                                                                                                                  | - | ٦. |
| S = 0.6 | $\sim$ |  |                                                                                                                  |   | 1  |
| 8.0 g H |        |  | and the second |   |    |
| 0.1 ast |        |  |                                                                                                                  |   | 7  |

Figure S2: Density contrast between the AW and the plume along the centreline of 79NG in the case where the AW consists of the warmest densest water found in the water column at 600 m depth.

4) Melt rate dependency

The authors discuss whether their melting sensitivity to ocean temperature is linear or nonlinear. I have several comments: i) The authors report a quadratic fit in Figure 5 but seemingly only based upon the 7 warmest temperatures. Why not use all of the temperatures? ii) They later claim that the fit is linear for the 7 warmest temperatures, which is fine, but that linear relation cannot be universally true since it does not pass through zero melting for zero thermal driving. So the nonlinear fit must be the more general relationship. iii) The linearity or otherwise is not rigorously tested using a statistical test. iv) On page 20 some reasons for the linearity are stated. As described above, I think the results are entirely consistent with the quadratic fits of Holland et al 2008 over the wider temperature range, and so there is no discrepancy to explain. Further, any discrepancy that is present would most obviously be explained by the lack of Coriolis force in the plume model.

i) The authors report a quadratic fit in Figure 5 but seemingly only based upon the 7 warmest temperatures. Why not use all of the temperatures?

We have implemented the suggested larger range for the temperature sensitivity in Figure 5. We did not do this earlier because we thought it was outside the interesting range.

Figure 5. Mean melt rate along the centreline of the 79NG ice tongue (blue), maximum melt rate in the grounding line zone (red), and melt rate calculated using the quadratic fit function (green) depending on the AW temperature as described in the text.

ii) ... linear relation cannot be universally true since it does not pass through zero melting for zero thermal driving. So the nonlinear fit must be the more general relationship.

This is a valid point. After performing statistical tests we conclude that the relationship is indeed nonlinear.

iii) The linearity or otherwise is not rigorously tested using a statistical test.

We agree that this is a useful addition to the sensitivity and have used several statistical tests as described below. This lead to some new related text that will be implemented.

We tested a linear model and a quadratic model using ANOVA test with a confidence interval of 95% in MATLAB (Table S1). The residual sum of squares for the linear model (93.7) is an order of magnitude larger than for the quadratic model (4.2), and thus, the quadratic model is a better fit for the melt-rate dependency. The linear part becomes larger for smaller temperatures. A quadratic dependency is also clearly revealed when plotting the raw residuals against the fitted melt rates (Figure S3, right). We conclude, that the dependency of the melt rate to the AW temperature is quadratic, though, with a linear part, also indicated in the fitting equations 9 and 10 (Figure 5). We thus agree that our results displayed in Figure 5 are entirely consistent with the quadratic fits of Holland et al., 2007. However, several other studies report a linear dependency of the melt rates to

ocean warming for different ice shelves (e.g., Williams et al., 2002; Rignot and Jacobs, 2002; Shepherd et al., 2004; Payne et al., 2007).

---

## Author Comment (AC2) · 9 Jul 2019

**Dear Anonymous Reviewer #2,**

Thanks for your invested time in reading our manuscript and for your many and detailed constructive suggestions. We hope that you will find the new and improved version of the paper satisfactory. Our response is written in bold font directly below your original comments in the text below. We have responded to all suggestions, followed most of them, and clarified the issue if required.

We will made an attempt to shorten the paper, but were also told by reviewer #1 to add some text on the tides and seasonal changes in subglacial discharge and AW temperature. These additions hopefully contribute to the novelty of our work.

Best regards,

Philipp Anhaus, Lars H. Smedsrud, Marius Årthun, and Fiammetta Straneo

**Major points**

1) Ice-tongue melt-time projection

The ice-tongue melt-time projection neglects all feedbacks in this glacier fjord, including ice dynamics (evolving ice flux across the grounding line, and shape of the cavity), iceberg calving, evolving ocean circulation and wind patterns (and sea ice, if relevant) outside the fjord. The projection also does not take into account, rotational effects, dimensionality of the problem, and temporal changes inside the fjord - I would expect at least seasonal AW temperature/thickness (and presence), and seasonally distributed subglacial discharge to play some role here. As a result, I am not sure the provided ice-tongue melt-time projection has much meaning. If not a coupled glacier-ocean model, I think at least a glacier model should be involved, to produce a somewhat more robust statement about the 79NG stability at present and in the future.

We agree that the melt-time estimates neglects a number of the processes listed above, and we only calculated this to illustrate the sensitivity to ocean temperature in an easy-to-understand way. The paragraph will be rewritten to make this more clear. We will include new runs that includes seasonality in subglacial discharge and AW temperature/ layer thickness. These are both highly uncertain. The subglacial discharge seasonal variation has an overall small contribution, and this is also the case with the seasonality in the AW temperature/ layer thickness as observed with an Ice Tethered Mooring (ITM, Figure S6).

The ITM was deployed on a 1.35 m thick ice floe in a rift 15 km up-glacier from the northern terminus of the 79NG ice tongue during the ARK-XXX/2 (PS100) cruise on the R/V Polarstern (Kanzow, 2017) on August 23, 2016 at 79° 41.0 N, 20° 20.9 W. Four Aquadopp single-point current profiler from Nortek AS (http://www.nortekusa.com/usa/products/acoustic-doppler-current-meters/aquadopp-current-meter-brochure/view) were attached to the mooring line at initial depths of about 165 m, 250 m, 370 m, and 500 m (https://www.whoi.edu/page.do?pid=154416). The measurements including temperature were averaged over 15 minutes. The data were collected and made available by the Ice-Tethered Profiler Program (Toole et al., 2011; Krishfield et al., 2008) based at the Woods Hole Oceanographic Institution (http://www.whoi.edu/itp).

2) New insights

The main focus of the manuscript is on assessing sensitivity of the plume model to its parameters - but I am not sure it provides any new insights or conclusions. I am also not sure it provides any new insights

on the processes driving submarine melting - but is it possible that I missed it - it would be helpful if the authors clarified what the contributions are and how they differ from previous studies. Here are a few studies (not referenced here) that have done this before and more exhaustively: Carroll et al 2015, and 2016, Sciascia 2013, Beckmann et al 2018. As it stands, I think the presented plume model application to 79NG is within the parameter range studied previously. If not, it would be good to clarify that. The main result (linear scaling of melt rate with AW temperature) is consistent with other studies in Greenland glaciers, and as mentioned even in the discussion here it maybe more of a property of the plume model itself, than anything else.

The main goal of the paper is to estimate the sensitivity of ocean driven melting of 79NG. But to find this sensitivity we had to find a realistic set of parameters. We agree that the main finding appears to be consistent with earlier studies, but this is a good thing, because there are so few observations available of 79NG. So, despite the limited observations, results appear sound. We have further found that melting is in-sensitive to a large range in subglacial discharge (from RACMO2.3p2), and that tides are likely too slow to contribute much to the melting. Overall the sensitivity to AW temperature (from ECCOv4) seems large, and the present AW presence already melts the 79NG quite effectively.

We are thankful for the suggested new references that we have investigated and some will be included in the new version of the manuscript. They largely describe comparable results, but for other areas, or in a more general way using parameters not specific for the 79NG configuration.

3) References

The manuscript is not very carefully referenced. Although there are a lot of references, the choices are sometimes quite arbitrary. Given this is primarily a sensitivity study of the plume model - it should be clear how the findings here differ from other (often more complete and insightful) sensitivity studies of the plume model. Modelling studies are at times used as references where one would expect a reference for observations.

We are sorry to learn that you find other studies more complete and full of insight, but hold that we have also found some interesting results that will be useful. We will update the citations in many places, and include the results on tides and seasonality in subglacial discharge and AW temperature/ layer thickness – so the novelty should be improved.

4) Structure

I feel the manuscript is written quite confusingly and could use a bit of reorganization. Description and interpretation are often mixed without a clear distinction. Although I have quite a few in-line comments and clarification suggestions below, I feel that for what the manuscript does, it could be half its length, and more to the point. A lot of the confusion arises because of the poor organization. Here are some suggestions for restructuring:

\*Include a background section (could be part of introduction) - where all relevant information about the region and glacier is summarized and refer to it whenever necessary, instead of giving background throughout the manuscript, sometimes repetitively, sometimes not at all.

Thanks for the constructive suggestions on improving the structure of our paper. We will try to follow your suggestions by moving the first paragraph of the results chapter which addresses the hydrography at the 79NG (Wilson and Straneo, 2015) to the background in the introduction. Otherwise the background in the introduction is fairly short and contains the relevant information. The rest is methods, new results, and discussion of our results. We will attempt to delete some details about the model parameters, and focus more on the results in the present version.

\*Clearly describe the experiment setup in the methods section, motivate and justify these experiment choices and clarify what you are trying to achieve. State all the assumptions in this section. At present, while there is discussion of sensitivity to some parameters in the end, there are many model choices for which there is no reasoning/explanation provided

These are indeed general and good rules to follow, and we will follow most of the specific suggestions. We will merge the content of Table 2 in Table 1 in the methods section in order to describe all model parameters in this section instead of having this separated. The "problem" arises here because some model "choices" are indeed results, and needed some discussion. What we have provided in the methods section is the model equations and the STANDARD values for the model parameters. These are our best choice values, and were used for the sensitivity of melt rates towards AW temperature – the main focus. Our best shot at the mean situation is presented in section 3.2 (STANDARD case), and the sensitivity to the ocean forcing in section 3.3 (ocean forcing sensitivity). The detailed testing of the parameters will be shifted to section 3.4 (parametrization). It is probably this section that is thought of as "model choices" – but these are truly unknown, and we had to do extensive testing to find values that we think are representative.

\*Describe the relevant part of the results (I think currently the result description is quite long, given it doesn't provide that much new insight)

Hopefully people have different interests, and some will find the results regarding plume speed, plume thickness, entrainment rates, drag coefficients, temperature evolution and topographic control for 79NG interesting. No such results are available for 79NG, until now. We agree that the melt rates are probably the most interesting though and some were already presented in Schaffer (2017) and Mayer et al. (2018) but not the extensive testing of the coefficients, the subglacial discharge range we will update now, and the tides we will now describe. The paper is quite compact, and only has 8 figures. Four of them present the melting results. We also have only 3 tables now.

\*Explain why the results (resulting melt rates in this case) are believable for the base case (present), and only then move on to the results for the future case.

We will add a direct comparison with satellite-based melt rates in the new version. However, these were cited in the beginning of the discussion section (4.1). In our view the results section should only present our results, and then these should be compared to other work in the discussion part. Perhaps the suggestion here is to merge results and discussion – but we think the normal and best way is to have them separated. We will now include direct comparison between our melt rates from the plume model and the satellite derived melt rates from Wilson et al. (2017) in Figure (4a). This was also suggested by reviewer #1. We do not do 'future' cases, we only present a sensitivity towards AW temperature.

\*To discuss the future warming scenarios, I think some sort of a model would be needed (see points 1)

We have used the ECCO simulations to extract variability in AW properties, and we have used the plume model to estimate the melt rates given a range in these, so we have used two models. Perhaps a glacial, or ice-shelf model is meant here, which we have not used. We will rewrite the "melt-time" part – to clarify that we are indeed not presenting realistic scenarios for future changes in shape of the 79NG.

In-line comments

Abstract:

L7 - decay of what? 20 km from the grounding line?

Will be changed to: "Melt rates drop rapidly to 6 m/yr within the first 10-20 km from the grounding line."

L10 - why is the melt rate sensitivity reported along a centreline when just a few lines above the melt rate is divided into three sections? Also, is this range of melt rate or melt rate increase?

We think that both the maximum and mean melt rates are worth mentioning in the abstract. This will be changed to: "The melt rates increase quadratic with rising AW temperature, and overall mean melting changes from 10 to 21 m/yr with the changes in ocean forcing."

L13 – In which way does the manuscript improve the understanding of processes driving submarine melting of marine-terminating glaciers around Greenland? This is the place to be specific.

**Agreed, will be changed to: "Our results show that submarine melting of the marine-terminating 79NG is sensitive to changes in ocean temperature."**

Intro:

P2:

L5 - general statement - doesn't need a reference.

**We will delete the reference here.**

L6 - Holland 2007 seems a more appropriate reference here, than this review paper.

**We will change the citation accordingly.**

L7 - Do you mean net mass loss increase? Also, this sentence seems to contradict a previous sentence (P1 L20-22: The enhanced mass loss is caused by increased surface melt, and retreat and speed up of marine-terminating glaciers (Enderlin et al., 2014))

**Yes, will be changed to: "The net mass loss increase of marine terminating glaciers is believed to be caused by increased melting at grounding line depth, leading to inward migration of the grounding line and accelerated glacial flow (Thomas et al., 2009).**

L8 - The increase of submarine melting.... not the submarine melting itself - ...leads to an inward migration..... A stable glacier can still have submarine melt.

**Yes, will be corrected as stated above.**

L9 - "It is important to study submarine melting since it is a likely trigger of change of ice loss from the ice sheet." Again, the presence of submarine melting is not a trigger of change, it is the change of submarine melt that may act as a trigger of change.

**Yes, correct. This will be changed to: "It is important to study submarine melting, because an increase could trigger increased loss of the ice sheet."**

L14 - buttressing is defined usually at the grounding line, not at the terminus, could you specify what you mean by buttressing at the terminus here?

**Suggestion will be followed: "at the terminus" will be deleted.**

L26 - Are these references supposed to refer to observations of melt/discharge driven plumes? Two of these are models, not observations

**Sentence will be rewritten, also suggested by reviewer #1.**

L27 – which of the two scenarios are likely to happen at NG? and why is it likely? The majority of the subglacial discharge is most likely released at depth —- are you referring to 79NG specifically here? again, why is it likely? is there any support for this, or is that an assumption (which is completely fine as long as it is clarified) based on observations elsewhere?

There is in general very few observations available for 79NG, and then we have used the word "likely" to state that things are "probably" similar here to other Greenland areas. Subglacial discharge is hard to observe in general, so this is also primarily unknown in general.

L29 - This is another awkward choice of reference. There have been plenty of earlier studies characterizing channelized network under ice tongues/ ice shelves.

Since Dallaston does not relate to 79NG or even Greenland specifically, I don't understand the choice for this particular reference here as opposed to earlier ones.

We found this to be a general good description. We will also cite Rignot and Steffen (2008) and Millgate et al. (2013) which both investigate channelized bottom melting at the floating ice tongue of Petermann Glacier in North Greenland based on observations (Rignot and Steffen, 2008) and the model MITgcm (Millgate et al., 2013).

L33 - Isn't it 50% calving and 50% submarine melt? The wording here suggests that 50% is from these two together and the remainder from something else.

**This addresses Greenland, so the remaining part is the surface melt.**

L34 - This manuscript really overuses the word likely. It would help to clarify what is known (reference), what is speculated (reference), and what is assumed for the purposes of this study.

**As noted above is there often no specific observation available for the 79NG. We agree that it is good not to use the same phrases, and we will use a different term where this is just as correct.**

P3:

L5 - In my view this is more of a sensitivity study, not a process study

Yes, we agree. This will be changed to: "The main goal of this study is to address the sensitivity of submarine melting of 79NG to changes in ocean forcing. However, to do this we also needed an improved understanding of the ocean processes below the ice tongue".

Data:

P3:

L20 - What is a high-res digital bathymetry model and how does it compensate poor data coverage?

Will be changed to: "The poor bathymetric data coverage ..." So what is meant here is that the additional CTD profiles, echo-soundings and model make the bathymetry better.

L18-23 - Bathymetry - did you do this data merging here for the purpose of this manuscript? If so this needs to be described in much more detail. If not, it would be good to first write what product is used in which part of the MS, followed by a brief description what this product consists of.

The plume model uses only the ice base data, but we also plot bedrock and ice elevation in the figures, so we actually use all of it. We will add this: "Bedrock, ice elevation, and ice base along the centreline of the 79NG ice tongue are shown in Figure 1 and spatially in Figure 2b, but it is only the ice base that is directly used by the plume model."

L24 - This third paragraph logically follows from the first one, not from the second one as it refers back to the ice, so perhaps rearrange. Also please explain the choice of the spatial filter, and how it guarantees an increase of the ice base, or if this was then enforced by some other procedure.

**The lines 24-27 will be moved up and merged with the first paragraph of the section.**

L25 - How were the plume paths chosen? If it is discussed later perhaps reference the section.

Will be changed to: "Data points were manually extracted and smoothed by a moving average (boxcar filter) every 5th data point along the three selected plume paths along the centerline, northern and southern sides of the fjord."

P4:

L1 - Which tidal effects? Do you include those here or you use the basic version?

Will be changed to: "Frazil ice growth and precipitation, as well as increased vertical shear from tidal currents was included by Smedsrud and Jenkins (2004). This was tested for the 79NG as well, but no super cooling is produced, and there is no frazil ice dynamics."

Tidal effects were investigated in Anhaus (2017, Master's thesis, unpublished). The mean tidal velocity in the cavity was estimated to be 1.18 cm/s using the current data collect by the ITM and the harmonic analysis package T\_tide (Pawlowicz et al., 2002). The period October 21, 2016 to January 18, 2017 was extracted which gives a record length of 89.3 days, sufficient to detect all tidal constituents.

The tidal flow of 1.18 cm/s is too weak to contribute effectively to the melting and plume dynamics (velocity and thickness). This was concluded from applying no tidal flow as well as adding the tides in the plume model, and the results were similar (Figure S1). Tides might be low in the cavity of 79NG because of ice blocking the flow. However, this explanation is speculative at best.

Figure S1: Sensitivity of the submarine melt rate along the centreline of the 79NG ice tongue due to the tidal flow in the cavity. The tidal velocity in the STANDARD case is 1.18 cm/s. Note that this are results from Anhaus (2017, Master's thesis, unpublished) and here the STANDARD case has tides of 1.18 cm/s and a subglacial discharge of 1 x 10e-03 m2/s.

Mortensen et al. (2014) performed a tidal analysis at Godthåbsfjord in West Greenland also using moored current meter measurements. Maximum tidal velocities were associated with the M2 and S2 component and 4 - 5 cm/s and 1 - 2 cm/s. The tidal flow in the cavity below the 79NG ice tongue is thus low compared to tidal velocities around Greenland. Moreover, tides are fairly barotropic

(Anhaus, 2017, Master's thesis, unpublished) and, thus, does not seem to influence the entrainment of AW.

In general, a stronger tidal flow would increase the shear between the plume and the ice-ocean boundary and, thus, the drag. This causes the plume to slow down and, as a result, less AW is entrained which lead to less melting. This response is supported by Smedsrud and Jenkins (2004) and investigated in Anhaus (2017, Master's thesis, unpublished) for the 79NG (Figure S1).

L5 - Why is STANDARD capitalized? At least at this point of the manuscript this is not at all clear.

We chose to capitalize STANDARD to make it clear that this is a model simulation and not the regular meaning of the word. We will add this: "We use the term STANDARD to name the set of values used in our regular model simulation, so the STANDARD value for subglacial discharge used is  $4.0 \times 10e-03 \text{ m}^2/\text{s}$  (Table 1).

L5-7 - This part is a bit rushed, could you be a bit clearer on how you derive the subglacial discharge, and what assumptions go into the derivation. Further, have you considered separating the summer and winter case? Presumably the subglacial discharge is very seasonal and unless the plume model depends linearly on this parameter, using a long term annual mean might over- or under-estimate the melt.

In an effort to produce a compact and easy to follow paper we postponed the details to section 3.4. We have done simulations to address changes in subglacial discharge, and will add some text to describe the seasonality (Figure S2). We also investigated the effect of having the subglacial discharge distributed uniformly along the grounding line (GL) and as one single source (narrow opening, NO). As it turns out the melting is quite independent of a large range of this forcing.

Figure S2: Sensitivity of the submarine melt rate along the centreline of the 79NG ice tongue due to the seasonality and distribution of subglacial discharge ranging from  $5.4 \times 10e-06 \text{ m}^2/\text{s}$  (winter, GL) to 0.298 m2/s (summer, NO). The STANDARD value for subglacial discharge used is the long-term annual mean 4.0 x 10e-03 m2/s from RACMO2.3p2.

L9 – I am not sure I am familiar with the terminology "line source equation" could you clarify what this term means?

**We will delete "line source" in the text, it is not required.**

L9-10 - it is the quantities, not the fluxes of the quantities that are conserved.

**Indeed correct, we will delete "fluxes of" here.**

P5: How is the "re-circulation" and "the southern and northern recirculation in Fram Strait" shown in the figure?

The re-circulation is the flow in Fram Strait that turns southwards, one branch in the north, and one in the south.

P6:

L7 - What value do you use for the ref. density? I don't see it in the table.

**The reference density of 1028 kg/m3 will be added to Table 1.**

L8 - Clarify that initial doesn't refer to initial condition as there is no time dependency in the equations.

Will be changed to: "The initial start temperature T0 and salinity S0 of the plume ..."

L9 - Why is T0 set to freezing point? Does the result depend on different values of T0? What is S0 set to?

This is meltwater flowing alongside the glacier, so it must be at the freezing point. Will be changed to: "Here, To is set to the freezing point, and SO is zero as this is pure meltwater".

L11 - Melt rates at 79NG are .... This statement is not specific to NG, or is it?

Will be changed to: "Melt rates at 79NG as most other ice shelves are expected to be highest close to the grounding line".

L24 - Do you have any reference that -15C is reasonable, or how sensitive is the result to that?

This is a small part of the heat budget, so we have chosen not to do specific tests on this forcing. It would be easy to do though, but it represents a mean surface temperature during snow accumulation.

General - I think it would be clearer if the plume model was first presented in general, and only after all the concepts are introduced, you can introduce specific choices for 79NG and justify how appropriate they are. Constantly switching between these two makes it very confusing.

This is indeed what we aimed to do; present the model and the normal parameters in this section. The values that needed specific testing and were unknown for 79NG are tested in section 3.4 and discussed in section 4.3 and 4.4.

P7:

Table 1 - the values used for the constant, where are they taken from? – reference

**Most values are the same as used by Jenkins (2011) unless otherwise stated. This will be added to the table caption now: "Physical constants (left) and output variables (right) of the 1D ISW plume model. Unless stated otherwise constants are the same as in Jenkins (2011)."**

L6 - Having determined that rotation is important in this fjord, how is it taken into account? As far as I am aware applying 1D plume model to 3 different paths, does not deal with rotation effects - but that is what line P7 L9-11 seem to suggest.

The results are quite similar for the southern and central path, indicating that the buoyant outflow leans up against the coast to the south over a large area. In the north melt rates are much lower, but

this is caused by the very flat profile (Figure 8d, e). We describe these differences at the end of section 3.3 and discuss them in section 4.4.

L8 - is 2-layer an assumption, or is it an approximation based on observations?

The two layer assumption is here just required for calculating the Rossby radius, this will be made clear now: "To calculate the first baroclinic radius of deformation R ... the stratification is assumed to be a two-layer system consisting of PW overlying AW with densities ...".

L9 - define f when it is introduced not several lines later

Agreed. The first line after the equation will be changed to: "f is the Coriolis parameter, and the required water column thicknesses are the upper H1 (90 m), lower H2 (470 m), and total H (560 m) values from the CTD profile in the rift."

L11 - The differences between ... it seems like this belongs more to discussion/results than here.

**This result justifies the use of mainly using one plume path for the sensitivity.**

L13 - Since there is no dependency, aren't all variables diagnostic?

**The paragraph will be rewritten in response also to reviewer #1.**

P8:

L4 - I don't understand this what ....with S instead of Sb refers to

**This refers to the equation in the text without a number for T\_b. The whole paragraph will be rewritten also in response to reviewer #1.**

L5 - can you show that ECCOv4 does a reasonable job in this region? are there any data to constrain it here, if so, how well does it match them, if not, why do you think this coarsely resolved model represents well the water masses relevant to your computation?

Yes – all available observations have been used to constrain the ECCOv4. ECCOv4 assimilates observations and thus simulate hydrography better than a "normal" model, and model drift might be avoided (Forget et al., 2015). However, data coverage is sparse, especially on the continental shelf of Northeast Greenland and at 79NG (Figure S3). The data constraints are Argo floats (IFREMER), CTD profiles (NODC, WOA09), and moorings.

In Anhaus (2017, Master's thesis, unpublished), ECCOv4 was evaluated with CTD profiles collected across Fram Strait (FS2016) and on the continental shelf of Northeast Greenland (Figure S4) and investigated annual and monthly (August and September) variations in AW temperature, salinity, and layer thickness. The CTD profiles were collected during the FS2016 cruise with R/V Lance organized by the Norwegian Polar Institute in August/September 2016 which the first-author participated.

ECCOv4 reproduces the location of the maximum temperatures in the surface layer offshore from Spitsbergen and the cooling of AW across Fram Strait is reflected (not shown). The modelled East Greenland Current (EGC) in the upper 100 m to 200 m is similar to the one observed during FS2016 (not shown). The large scale eddy observed during FS2016 offshore the Continental Shelf (not shown) is not found in ECCOv4. It is possible that eddies are not resolved in ECCOv4 due to the coarse